# Dynamic regulation of canonical TGFβ signalling by endothelial transcription factor ERG protects from liver fibrogenesis

Neil P. Dufton[1], Claire R. Peghaire[1], Lourdes Osuna-Almagro[1], Claudio Raimondi[1], Viktoria Kalna[1], Abhishek Chauhan [2], Gwilym Webb [2], Youwen Yang[1], Graeme M. Birdsey [1], Patricia Lalor[2], Justin C. Mason[1], David H. Adams[2] & Anna M. Randi[1]

The role of the endothelium in protecting from chronic liver disease and TGFβ-mediated fibrosis remains unclear. Here we describe how the endothelial transcription factor ETS-related gene (ERG) promotes liver homoeostasis by controlling canonical TGFβ-SMAD signalling, driving the SMAD1 pathway while repressing SMAD3 activity. Molecular analysis shows that ERG binds to SMAD3, restricting its access to DNA. Ablation of ERG expression results in endothelial-to-mesenchymal transition (EndMT) and spontaneous liver fibrogenesis in EC-specific constitutive hemi-deficient ($Erg^{cEC-Het}$) and inducible homozygous deficient mice ($Erg^{iEC-KO}$), in a SMAD3-dependent manner. Acute administration of the TNF-α inhibitor etanercept inhibits carbon tetrachloride ($CCL_4$)-induced fibrogenesis in an ERG-dependent manner in mice. Decreased ERG expression also correlates with EndMT in tissues from patients with end-stage liver fibrosis. These studies identify a pathogenic mechanism where loss of ERG causes endothelial-dependent liver fibrogenesis via regulation of SMAD2/3. Moreover, ERG represents a promising candidate biomarker for assessing EndMT in liver disease.

[1] Vascular Sciences, Imperial Centre for Translational and Experimental Medicine, National Heart and Lung Institute, Imperial College London, London W12 0NN, UK. [2] Centre for Liver Research, Institute of Biomedical Research, Institute of Immunology and Immunotherapy, College of Medical and Dental Sciences, University of Birmingham, Birmingham B15 2TT, UK. Correspondence and requests for materials should be addressed to A.M.R. (email: a.randi@imperial.ac.uk)

Chronic liver disease (CLD) is an increasing global health burden; in the UK, liver disease is the fifth biggest cause of mortality with rates doubling from 1991 to 2007[1]. CLD can be triggered by numerous factors including diet, alcohol, viral infection and genetic disorders, which share common features including excessive inflammation[2], dysregulated transforming growth factor (TGFβ) signalling[3] and dramatic disruption in the vascular architecture of the liver[4, 5]. Recent studies have revealed that disruption of endothelial cell (EC) homoeostasis can initiate tissue damage and fibrosis[6–8]. Notably, EC have been shown to lose their lineage-specific markers and morphology and acquire a mesenchymal-like phenotype in a process termed endothelial-to-mesenchymal transition (EndMT)[9]. EndMT is associated with human pathologies such as early vein-graft rejection[10] and atherosclerosis[11], where it correlates with disease severity. EndMT has been shown to be induced by TGFβ in vitro and in vivo[10, 12], and to be enhanced by inflammatory mediators including TNF-α[13, 14].

Canonical TGFβ/bone morphogenetic protein (BMP) signalling can activate two opposing signalling cascades, maintaining homoeostasis via phosphorylation of transcription factors SMAD1/5/8, while instigating pro-fibrotic signalling via phosphorylation of SMAD2/3[15] (Supplementary Fig. 1). In EC, TGFβ isoforms activate SMAD1/5/8 signalling via the receptor ACVRL1 and its co-factor endoglin (ENG)[16]. BMP ligand-BMP receptor (BMPR) and TGF-β-ACVRL1 interactions selectively induce SMAD1 phosphorylation and directly inhibit TGFβ-ALK5-SMAD3-mediated transcription[17]. Importantly, TGF-β-induced signalling is influenced by cross-talk with multiple pathways and by lineage-specific co-factors[18]. Thus, regulation of the balance between SMAD1 and SMAD3 signalling is crucial in maintaining EC homeostasis[19, 20].

The ETS transcription factor family plays important roles in vascular development and angiogenesis[21]. The ETS-related gene (ERG) is the most abundant ETS factor in adult ECs and is essential for endothelial lineage identity; indeed, it is one of only three transcription factors required for reprogramming of progenitors to endothelium[22]. ERG is crucial for embryonic development and vascular stability[23, 24], angiogenesis[25] and protection from vascular inflammation[26, 27]. ERG exerts its pro-homoeostatic, anti-inflammatory function in ECs by driving expression of homoeostatic genes while repressing pro-inflammatory gene expression and inhibiting cytokine-induced EC activation[21, 28]. In turn, pro-inflammatory molecules TNF-α[26] and lipopolysaccharides (LPS)[27] induce a significant loss in ERG expression in ECs, suggesting that regulation of ERG expression is key to control of the balance between endothelial homoeostasis and inflammatory signalling.

In this study, we show that ERG maintains the homoeostatic balance of SMAD-dependent signalling in the endothelium, by promoting the SMAD1 pathway while repressing SMAD2/3 activity. We show that loss of ERG in vitro and in vivo results in spontaneous EndMT which is dependent on enhanced SMAD2/3 activity. TNF-α blockade protects ERG expression and reduces phosphorylation of SMAD2/3 in an acute model of liver fibrosis, but is ineffective in ERG hemi-deficient mice. Finally, we show that ERG expression is lost in liver EC from cirrhotic patients with fibrosis related to alcoholic liver disease (ALD) or primary biliary cirrhosis (PBC) and inversely correlates with increased markers of EndMT. Therefore, this study identifies a central role for ERG in regulating EC canonical TGFβ-SMAD signalling to prevent EndMT and ultimately tissue fibrogenesis. Loss of endothelial ERG expression is an early, causative event during tissue fibrogenesis, linked to inflammatory pathways, which can be targeted therapeutically.

## Results

**ERG regulates SMAD signalling in EC and protects from EndMT.** Gene ontology analysis of transcriptome data from ERG-deficient human umbilical cord endothelial cells (HUVEC)[25] identified 41 target genes associated with TGFβ/BMP canonical signalling as putative ERG targets (Supplementary Table 1; Fig. 1a). qRT-PCR validation of selected hits and other relevant genes is show in Fig. 1b. Inhibition of ERG in HUVEC caused a reduction in gene expression of *SMAD1*, *ACVRL1* (*ALK1*), endoglin (*ENG*) and inhibitor of DNA binding 1 (*ID1*). This response was mirrored by elevated expression of *TGFβ* isoforms 1 and 2 and of mesenchymal genes collagen A1 (*Col1A1*), smooth muscle actin (*SMA*) and smooth muscle calponin (*CNN1*). Gene expression levels of other signalling SMADs (5 and 8) were not significantly changed in *ERG*-deficient cells (Supplementary Fig. 2A). A second siRNA (#2) targeting ERG generated a similar mRNA profile (Supplementary Fig. 2B). Protein levels of SMAD1 and its associated proteins, ENG and ID1, were also significantly reduced (Fig. 1c, d and Supplementary Fig. 2C). Loss of SMAD1 expression was confirmed in vivo, within liver tissue from EC-specific constitutive *Erg* hemi-deficient mice[23] (*Erg^{cEC-Het}*; Fig. 1e; arrows). Gene and protein expression levels of SMAD3 and its receptor TGFBR1/ALK5 were unaffected (Fig. 1b–d). However, there was increased expression of the downstream targets SMA (Fig. 1d) and TGFβ2 (Fig. 1f). Furthermore, immunofluorescence microscopy revealed increased TGFβ2 expression in ERG-deficient EC in vitro (Fig. 1g) and in large blood vessels in the liver of *Erg^{cEC-Het}* mice in vivo (Fig. 1h; arrows). The profile of ERG-deficient cells, namely marked upregulation of mesenchymal markers and downregulation of endothelial lineage identity markers, together with transition from the typical rounded/cobblestone morphology to a distinct mesenchymal-like shape[25], suggest that the cells are undergoing EndMT. Thus, ERG reciprocally regulates TGFβ/BMP canonical signalling in EC, by driving the SMAD1 pathway while repressing the SMAD2/3 pathway to protect EC from EndMT.

**ERG promotes SMAD1 whilst repressing SMAD2/3 signalling in EC.** To assess whether ERG inversely regulates SMAD1 and SMAD2/3 transcriptional activity, we used a SMAD1-dependent BMP Response Element luciferase reporter (BRE reporter)[29] and a SMAD2/3-dependent SMAD-Binding Element luciferase reporter (SBE reporter)[30]. The SMAD2/3 activator TGFβ2 induced enhanced transactivation activity of the SBE reporter (Supplementary Fig. 3A), with comparable effects at 1 and 10 ng ml⁻¹ TGFβ2. All following experiments were carried out with 10 ng ml⁻¹ TGFβ2, in line with previous in vitro studies on EndMT[31, 32]. Ligand specificity of BMP9 (1 ng ml⁻¹) and TGFβ2 (10 ng ml⁻¹) was shown by selective induction of their target genes *ENG* and *CNN1*, respectively (Supplementary Fig. 3B). In control siRNA transfected cells, treatment with BMP9-induced a robust BRE-, but no SBE-mediated signal, whilst TGFβ2 induced the opposite response, as expected (Fig. 2a, b). Inhibition of ERG expression caused a significant reduction in SMAD1-dependent BRE transactivation following BMP9 treatment (Fig. 2a); this was paralleled by decreased phosphorylation of SMAD1 in response to BMP9 (Supplementary Fig. 3C) and TGFβ2 (Supplementary Fig. 3D; quantified Supplementary Fig. 3E), suggesting that ERG regulates SMAD1 levels and activity. Notably, ERG overexpression did not further enhance BMP9-induced BRE reporter transactivation activity (Fig. 2c). TGFβ2 induced a modest increase in SBE reporter transactivation, which was significantly enhanced following the deletion of ERG by siRNA (Fig. 2b). Conversely, ERG-overexpression completely inhibited TGFβ2-

induced SBE reporter transactivation (Fig. 2d). Basal phosphorylation of SMAD3 was increased in ERG-deficient HUVEC compared to control (Supplementary Fig. 3D, F, G). These data indicate that ERG promotes SMAD1 expression and activity, while repressing SMAD2/3 activity through a non-transcriptional mechanism (Fig. 2e).

To investigate whether ERG modulation of SMAD2/3 activity was mediated by the formation of a regulatory complex, we assessed protein interaction by co-Immunoprecipitation (Co-IP)

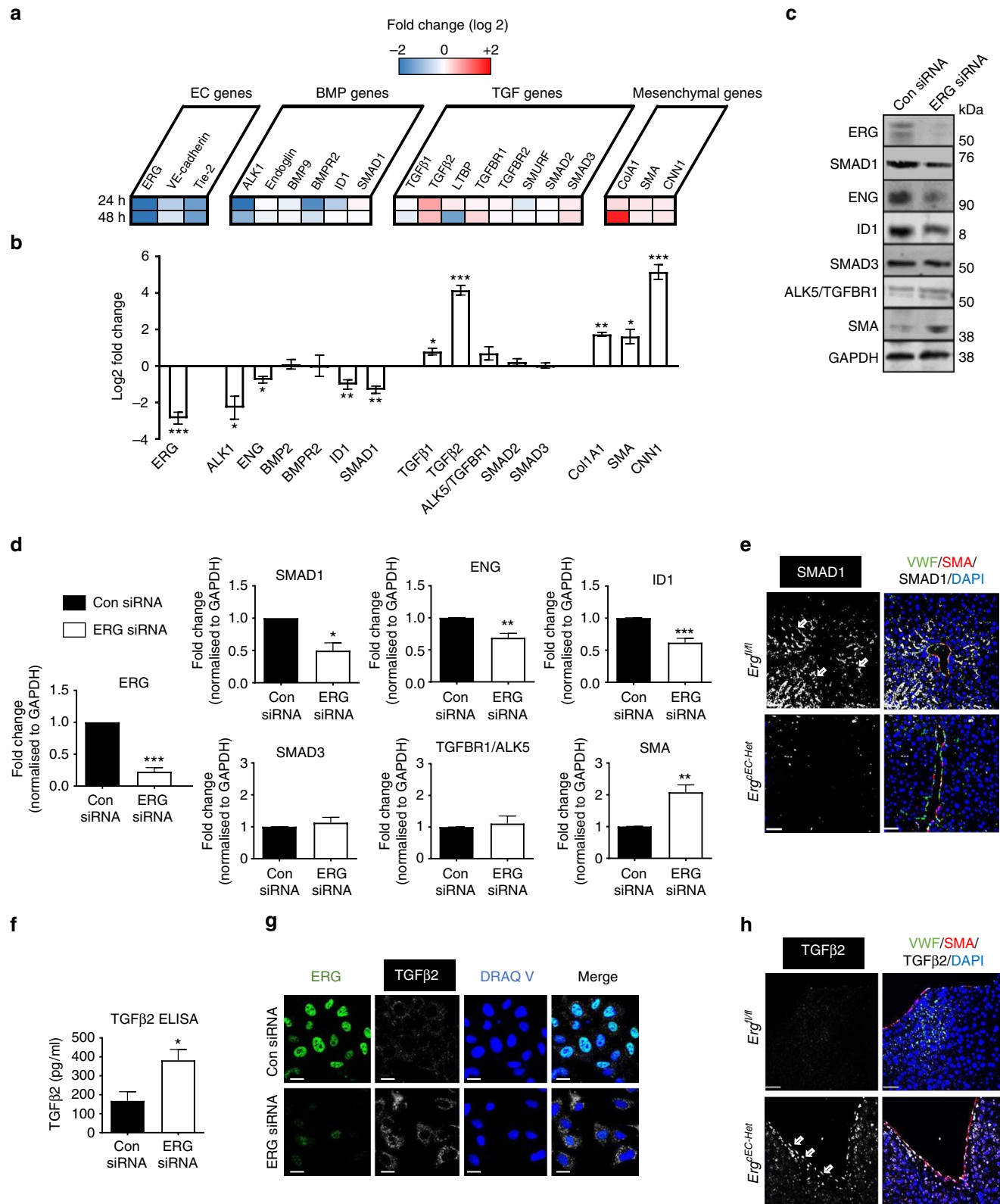

and proximity ligation assay (PLA). Co-IP revealed that ERG interacts with SMAD3 in HUVEC (Fig. 2f, Supplementary Figs. 4A, 5B) and in Hepatic Sinusoidal Endothelial Cells (HSEC; Fig. 2g) isolated from transplant patients with end-stage liver disease. Interestingly, ERG was found to also bind to SMAD2 (Fig. 2f), while no complex formation was observed with SMAD1 in either cell type (Fig. 2f and Supplementary Fig. 4A). TGFβ2 treatment resulted in the phosphorylation and translocation of SMAD3 into the nucleus of both HUVEC (Supplementary Fig. 4B) and HSEC (Supplementary Fig. 4C). This was paralleled by increased ERG/pSMAD3 complex formation assessed by Co-IP (Fig. 2f; Supplementary Fig. 4A). PLA confirmed that TGFβ2 treatment induces ERG-pSMAD3 interaction within the nucleus of HUVEC (Fig. 2h) and ERG-SMAD3 in both cell types (Supplementary Fig. 4D); no co-localisation was observed between ERG and SMAD1 (Supplementary Fig. 4D; controls Supplementary Fig. 4E). Inhibition of SMAD2/3 phosphorylation by pretreatment with SB-431542 prevented ERG-SMAD3 complex formation (Fig. 2f), suggesting that this interaction is dependent on SMAD3 phosphorylation. Selective removal of nucleic acids with benzonase treatment from the protein lysates (Supplementary Fig. 5A) did not significantly de-stabilise this interaction (Supplementary Fig. 5B), suggesting that the ERG-SMAD3 interaction can be maintained in the absence of DNA. Notably, SB-431542 normalised the enhanced expression of SMAD3-target genes TGFβ2 and CNN1 in ERG-deficient HUVEC (Fig. 2i) and HSEC (Fig. 2j), indicating that the EndMT phenotype in ERG-deficient EC is dependent on SMAD2/3 activity. Thus, these data demonstrate that ERG inhibits SMAD2/3 activity by direct interaction and formation of an inhibitory complex.

**ERG regulates SMAD3-DNA binding to repress SMAD3 activity.** Bio-informatic analysis of ERG-repressed-SMAD2/3-driven target genes TGFβ2 and CNN1 revealed the presence of highly conserved ERG DNA binding motifs, upstream of the transcription start site (TSS), which aligned with histone marks for active promoter regions, namely H3K4Me3, H3K27Ac, H3K9Ac and RNA polymerase II (RNA Pol2) occupancy [from Encyclopaedia of DNA Elements (ENCODE)] (Fig. 3a, b). We investigated binding of ERG and SMAD3 to these regions, which also contain several SMAD consensus motifs (Supplementary Fig. 6A, B). Chromatin immunoprecipitation (ChIP)-qPCR showed that in unstimulated HUVEC both ERG (Fig. 3c) and SMAD3 (Fig. 3d) are significantly enriched on the promoters of TGFβ2 and CNN1. TGFβ2 stimulation significantly reduced ERG enrichment (Fig. 3c) with concurrent increase in SMAD3 enrichment (Fig. 3d). ERG siRNA treatment led to the loss of ERG enrichment, as expected (Fig. 3e) and resulted in a significant increase in the binding of SMAD3 to the promoters of both target genes (Fig. 3f). These data indicate that ERG inhibits SMAD3 binding to DNA at these target gene promoters (model in Supplementary Fig. 6C).

**Erg-deficient mice undergo EndMT and liver fibrogenesis.** To determine whether the loss of endothelial ERG expression could influence tissue fibrogenesis in vivo, we assessed tissues from EC-specific constitutive Erg hemi-deficient mice (ErgcEC-Het)[23] and EC-specific inducible Pdgfb-iCreER-eGFP/Ergfl/fl mice (ErgiEC-KO; genotype data in Supplementary Fig. 7A, B). Increased collagen deposition and elevated SMA expression was most pronounced in the liver of both transgenic lines (Fig. 4a). We observed that reduction or deletion of EC-Erg in both strains caused disrupted portal tracts (schematic Fig. 4b), with significantly increased peri-portal collagen deposition (Fig. 4c) and SMA expression (Fig. 4d; quantification Fig. 4e, f; each genotype was compared to Ergfl/fl). Lineage tracing in ErgiEC-KO mice revealed eGFP+SMA+ EC (Fig. 4g, arrow), a sign of spontaneous EndMT, which was confirmed by quantification of CD31+SMA+ double positive cells (representative image Supplementary Fig. 7C, open arrows; quantified in Fig. 4h).

Activation of SMAD3 signalling was observed in the liver endothelium in Erg-deficient lines, corresponding to the in vitro results. pSMAD3 staining, undetectable in littermate control samples, was increased in both ErgcEC-het and ErgiEC-KO mice (Fig. 5a, b), in EC as well as surrounding tissue. This was accompanied by proliferation of biliary cells, detected by Ki67 expression, a sign of tissue dysfunction (Supplementary Fig. 7C, D). These data suggest that loss of EC-ERG induces both autocrine and paracrine responses through SMAD3 activation. In parallel with the in vitro studies (Fig. 2i, j), systemic administration of the ALK5 inhibitor SB-431542 abolished spontaneous SMAD3 phosphorylation in ErgcEC-het mice (Supplementary Fig. 8A). Furthermore, in vivo SB-431542 treatment normalised TGFβ2 expression in isolated primary mouse EC (Fig. 5c) and normalised both SMA expression (Fig. 5d, e) and collagen deposition (Fig. 5f and Supplementary Fig. 8B) compared with vehicle (DMSO)-treatment in ErgiEC-KO mice. These data show that loss of endothelial ERG expression causes enhanced SMAD3 activity in both EC and surrounding tissue, resulting in spontaneous EndMT and a pro-fibrotic microenvironment within the liver.

**Loss of ERG in murine and human fibrotic liver disease.** The finding of spontaneous EndMT and fibrogenesis in Erg-deficient mice led us to investigate whether endothelial ERG plays a role in the pathophysiology of liver fibrosis, using a carbon tetrachloride (CCL4)-induced mouse model. Chronic administration of CCL4 for 8 weeks resulted in characteristic bridging fibrosis (Supplementary Fig. 9A), increased SMA (Fig. 6a, b) and pSMAD3 expression (Supplementary Fig. 9B) in peri-portal areas. Chronic CCL4 treatment caused loss of ERG expression in all EC, which was partially restored following an additional 3-week recovery period (Fig. 6a, c). ERG expression is downregulated by inflammatory stimuli such as TNF-α[26] and LPS[27]. As elevated inflammatory cytokines are associated with fibrogenesis, we assessed the impact of TNF-α inhibition on ERG expression (Fig. 6d) and

**Fig. 1** Differentially expression of canonical TGFβ/BMP-SMAD genes in ERG-deficient HUVEC. **a** Microarray analysis of ERG-dependent genes in HUVEC was performed at 24 and 48 h after ERG depletion, as described (n = 3 biological replicates)[24]; fold change (log 2) of selected TGFβ/BMP associated genes represented as high (red) and low (blue) expression compared to the median (white). Gene expression data were validated in HUVEC transfected with Control (Con siRNA) or ERG siRNA by **b** quantitative PCR and **c** immunoblotting, showing reduction of ERG protein levels following siRNA, **d** quantification by fluorescence intensity normalised to GAPDH for ERG, SMAD1, ENG, ID1, SMAD3, ALK5 and SMA (n = 5). **e** Representative images of SMAD1 expression (white; arrows identify expression in the sinusoidal endothelium), VWF (green), SMA (red) and DAPI (blue) in liver tissue from Ergfl/fl and ErgcEC-het mice (Scale bar 50 μm). **f** Quantitative analysis of TGFβ2 protein expression was performed by ELISA in whole cell HUVEC lysates (n = 7). **g** Representative image of TGFβ2 expression in HUVEC transfected with control or ERG siRNA by immunofluorescence; nuclei are identified by DRAQ V and cells are co-stained for ERG (green). Scale bar 20 μm. **h** Representative images of TGFβ2 expression (white; arrows) in large vessel within the liver from Ergfl/fl and ErgcEC-het mice (Scale bar 50 μm). Data were normalised to GAPDH and compared to control siRNA treated (*) by unpaired t-test. All graphical data are mean ± s.e.m., *P < 0.05, **P < 0.01, ***P < 0.001

SMAD3 activity (Fig. 6e) by co-administration of the TNF-α antagonist etanercept in an acute CCL$_4$-induced liver injury. A single CCL$_4$ injection caused a significant increase in SMA (Fig. 6f), reduction of ERG expression (Fig. 6g) and enhanced phosphorylation of SMAD3 at 48 h (Fig. 6h and Supplementary Fig. 9B); co-administration of etanercept significantly reduced CCL$_4$-induced SMA expression (Fig. 6f), normalised endothelial ERG expression (Fig. 6g) and reduced phosphorylation of SMAD3 (Fig. 6h and Supplementary Fig. 9C). However, etanercept was unable to correct the CCL4-induced injury in $Erg^{cEC-het}$ mice (Fig. 6i), with no normalisation of transcriptional target genes (Fig. 6j) or SMA expression in the liver (Fig. 6k).

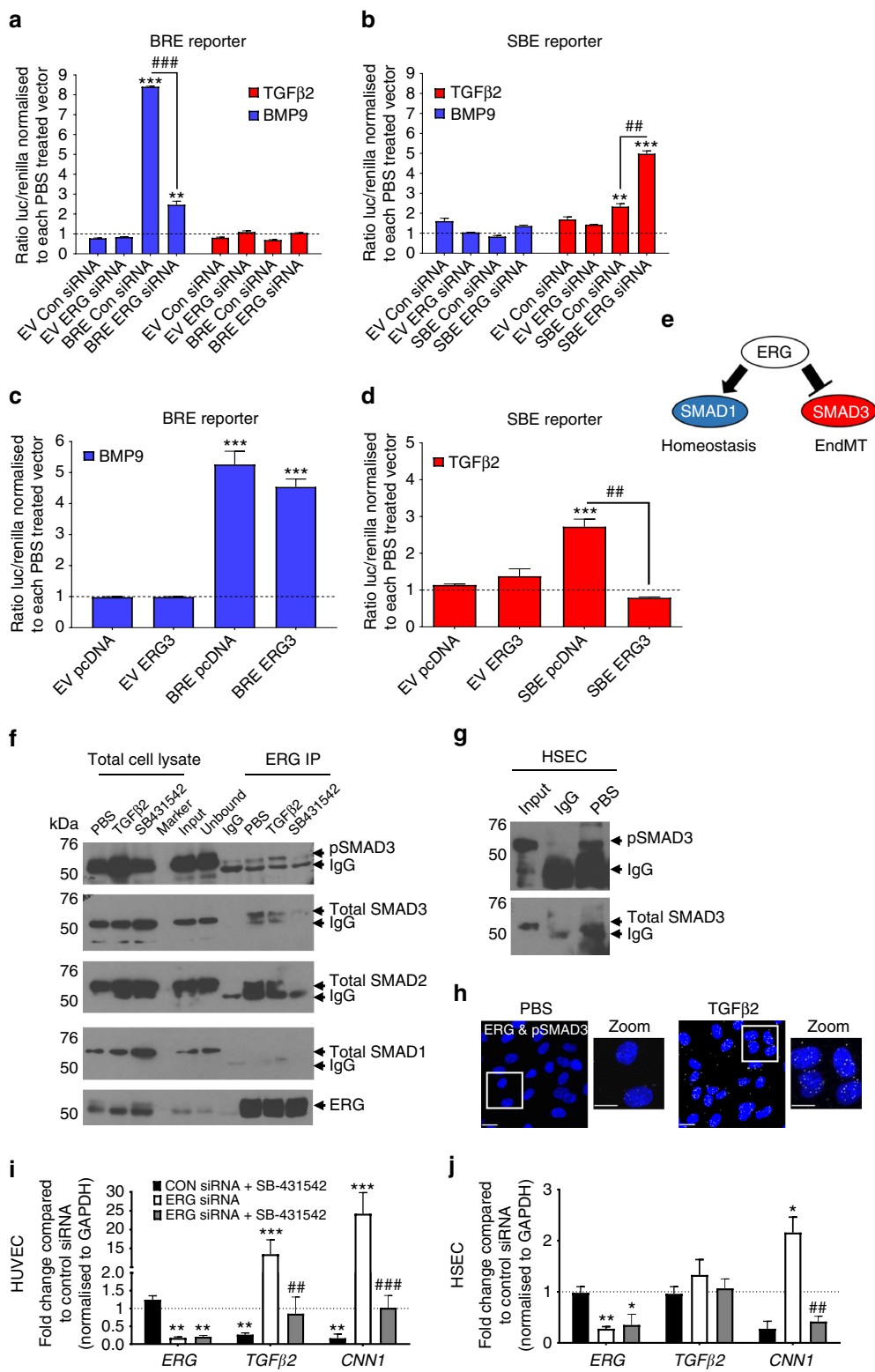

These data suggest that the TNF-α antagonist protects from CCL$_4$-induced tissue injury through the endothelial ERG pathway, and reveal that ERG expression is an early, sensitive marker of vascular inflammation during liver fibrogenesis, which can be modulated by clinically relevant therapies.

Finally, to verify the relevance of this pathway in human diseases associated with fibrosis, we obtained liver tissue samples from end-stage liver transplant patients with alcoholic liver disease (ALD), non-alcoholic steato-hepatitis (NASH) or primary biliary cirrhosis (PBC); non-fibrotic tissue from separate donors was used as control. Liver sections were profiled by immuno-fluorescence microscopy for CD31, SMA and ERG expression (Fig. 7a). In non-fibrotic human liver samples, ERG was expressed in both vascular and sinusoidal EC, with characteristic nuclear localisation (Supplementary Fig. 10A, B). ERG expression was significantly decreased in CD31$^+$ EC from ALD and PBC patient samples compared to non-fibrotic liver samples; in samples from NASH patient the decrease in EC-ERG expression did not reach significance (Fig. 7b; additional images provided in Supplementary Fig. 11A–D). EndMT, identified by CD31$^+$SMA$^+$ double positive cells, was identified in liver fibrosis associated with ALD and PBC but not in NASH patients (Fig. 7c). Analysis of CD31$^+$SMA$^+$ co-localisation with ERG expression in all patient samples revealed a significant negative correlation between ERG expression and EndMT (Fig. 7d). Therefore, loss of ERG expression appears to be a sensitive marker of inflammatory-driven fibrogenesis and could represent a novel biomarker for EndMT in human fibrotic liver.

## Discussion

In this study, we report that the endothelial transcription factor ERG controls the homoeostatic balance of TGFβ/BMP canonical signalling in vitro and in vivo; loss of endothelial ERG results in EndMT and spontaneous liver fibrosis in mouse models and inversely correlates with EndMT and liver fibrosis in patients.

Expression profiling of ERG-deficient HUVEC identified dysregulation of genes involved in SMAD1 and SMAD2/3 signalling. Expression of EC-restricted genes *SMAD1* and its co-receptors *ACVRL1* and *ENG* was reduced in ERG-deficient HUVEC, in line with the previous report of *ENG* as a direct ERG target[33]. Mutations in *ACVRL1* and *ENG* cause hereditary haemorrhagic telangiectasia (HHT), a disease characterised by vascular malformations[34, 35], in agreement with the model that this arm of the BMP/TGFβ pathway is required for vascular homoeostasis. Our data reveal that loss of ERG expression significantly affects SMAD1 phosphorylation and transcriptional activity. Co-IP and PLA assays revealed that ERG does not interact with SMAD1, suggesting that ERG is primarily required for the expression of key proteins in the SMAD1 pathway.

In ERG-deficient EC, loss of SMAD1-dependent transcription was accompanied by enhanced expression of SMAD2/3 target genes, indicating an inverse relationship between ERG and SMAD2/3-dependent targets. This shift in SMAD signalling is consistent with an EndMT phenotype, characterised by the loss of EC identity and the emergence of a mesenchymal gene expression profile[9, 10, 13]. We demonstrate that ERG directly represses SMAD2/3 transcriptional activity, independently of expression of SMAD2/3 or its receptor ALK5. We find that ERG can interact with SMAD2/3 in HUVEC and in HSEC in agreement with studies that show overexpression of exogenous ERG and SMAD3 results in protein interaction[36, 37]. ERG and SMAD3 both bind to the promoters of target genes, indicating that ERG acts as a co-repressor for SMAD3. These data are in line with a study mapping SMAD2/3 global DNA binding in keratinocytes, which revealed that ETS motifs are represented in ~50% of SMAD-enriched genes[38], suggesting that ETS/SMAD dual motifs may be important in the regulation of TGFβ/SMAD signalling.

Previous work investigating the regulation of TGFβ/BMP signalling by ERG in vivo has focused on developmental biology. ERG was shown to be involved in endocardial-mesenchymal transition in cardiac valve morphogenesis[39] and for TGFβ-mediated differentiation of sclerotome cells during vertebrae bone development[36]. Here we utilised two EC-specific *Erg*-deficient mouse strains to assess the role of ERG in regulating SMAD signalling in post-natal liver physiology; this is the first study to identify ERG expression in liver vascular and sinusoidal EC. Both strains displayed deformation of portal tracts with increased collagen deposition, SMA and pSMAD3 expression confirming a pro-fibrotic phenotype. Furthermore, analysis of CD31$^+$SMA$^+$ double positive EC and lineage tracing indicates that *Erg*-deficient EC undergo spontaneous EndMT in vivo, which was dependent on SMAD3, as shown by rescue of the phenotypes with SB-431542. Interestingly, endothelial ERG deficiency caused enhanced pSMAD3 not only in the endothelium, but also in the surrounding tissue. Thus, these data suggest that ERG is central to EC-specific regulation of SMAD2/3 and promotes tissue homoeostasis by preventing a paracrine TGFβ2-SMAD3 positive feedback loop, resulting in a pro-fibrotic tissue microenvironment.

In a murine model of acute and chronic CCL$_4$-induced liver fibrosis, we observed a rapid, marked and persistent reduction in ERG expression. Notably, ERG expression was partially restored in recovering tissue (3 weeks after terminating chronic CCL$_4$ administration) and protected by co-administration of the TNF-α inhibitor etanercept. By profiling ERG expression in human liver, we found marked loss of ERG expression in end-stage liver tissues from ALD and PBC patients. These diseases have been associated with inflammatory pathways[40, 41]. Loss of ERG expression could be due to chronic vascular inflammation; a similar observation

**Fig. 2** ERG differentially regulates transcriptional activity of SMAD1 and SMAD3. ERG-dependent regulation of SMAD1 and SMAD3 activity were assessed by transactivation assays using either SMAD1 reporter pGL3-BRE (BRE reporter, **a** and **c**) or SMAD3 reporter pBV-SBE4 (SBE reporter, **b** and **d**). HUVEC were either co-transfected with Control siRNA (Con siRNA) or ERG siRNA for 24 h (**a** and **b**) or with pcDNA or ERG3 overexpression construct (ERG3) (**c** and **d**) prior to treatment with TGFβ2 (red) and BMP9 (blue) or PBS (dashed line) for 18 h (data from pooled HUVEC in triplicate experiments). The ratio of luciferase to renilla from each transfection was normalised to PBS-treated control or to groups co-transfected with (**a** and **b**) Con siRNA or (**c** and **d**) pcDNA (n = 3). **e** Schematic of the inverse regulation of SMAD1 and SMAD3 by ERG. **f** Protein−protein interactions between ERG, SMAD1, SMAD2 and SMAD3 were assessed by Co-IP assay in whole cell lysate from HUVEC. SB-431542 (SB; 10 μM) treatment was performed for 1 h and TGFβ2 (10 ng ml$^{-1}$) or PBS treatments were performed 30 min prior to lysis. Lysates were immunoprecipitated with mouse IgG or mouse α-ERG and then immuno-blotted for α-ERG, α-SMAD1, α-SMAD2, α-SMAD3 and α-pSMAD3 (Ser423/425). Images representative of four experiments. **g** ERG-SMAD3 interaction was assessed by Co-IP in untreated HSEC. **h** Cellular localisation of the interaction between ERG with pSMAD3 was investigated by Proximity Ligation Assay (PLA) following 30 min TGFβ2 treatment in HUVEC (n = 2). **i** HUVEC or **j** HSEC were pre-treated with SB-431542 or DMSO prior to transfection with either control siRNA or ERG siRNA for 48 h and analysed by qPCR (n = 3). Data were normalised to GAPDH and compared to control siRNA treated (*) or to ERG siRNA treated with DMSO (#) by unpaired t-test. All graphical data are mean ± s.e.m., * or #P < 0.05, ** or ##P < 0.01, *** or ###P < 0.001

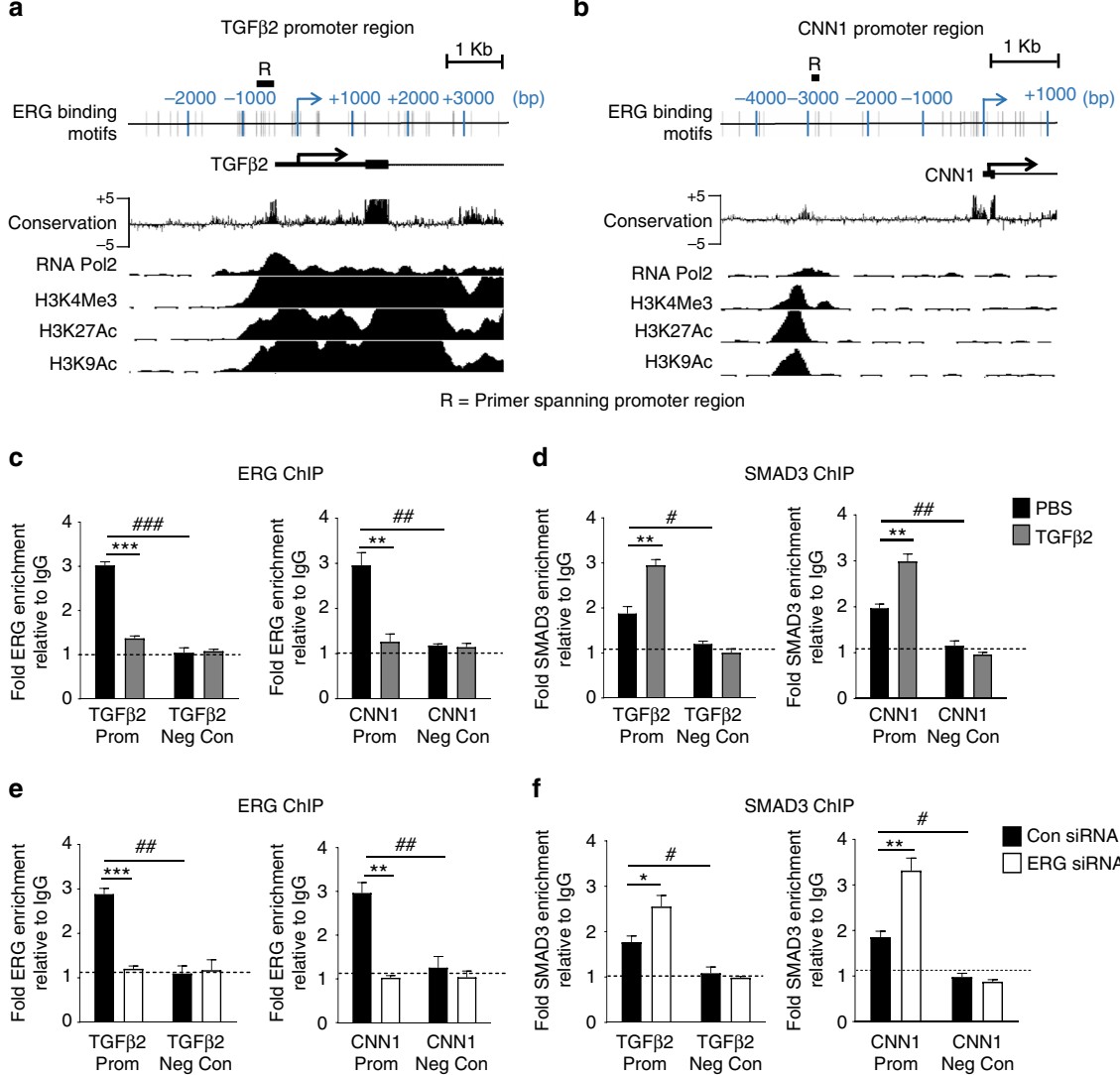

**Fig. 3** ERG is a key factor for endothelial specific inhibition of SMAD3-DNA binding. Putative ERG binding sites (grey bars) are located within the **a** TGβ2 and **b** CNN1 promoters upstream of the transcription start site (TSS) (arrow); ENCODE sequence conservation between 100 vertebrates across this region is shown. ENCODE ChIP-seq data profile for H3K4Me3, H3K27Ac, H3K9Ac and RNA polymerase II (RNA Pol2) in HUVEC indicate open chromatin and active transcription. Location of qPCR amplicon region R is indicated. ChIP-qPCR analysis of HUVEC was assessed following treatment with **c** and **d** TGFβ2 or PBS for 30 min, or **e** and **f** control or ERG siRNA. Chromatin was immunoprecipitated with **c** and **e** an α-ERG antibody, **d** and **f** α-SMAD3 antibody or control IgG. DNA was analysed by qPCR comparing specific primers against a negative control region for each target. Results are expressed as fold change compared to IgG ($n = 3$ for all experiments). Basal enrichment was analysed compared to negative control region (#). TGFβ2 or ERG siRNA treatment were compared to PBS treated or control siRNA (*) mean ± s.e.m., * or [#]$P < 0.05$, ** or [##]$P < 0.01$, *** or [###]$P < 0.001$ by unpaired $t$-test

was reported in the endothelium overlaying human atherosclerotic plaques[26]. Anti-TNF therapy has been effective in preclinical liver fibrosis models[42, 43]; furthermore, etanercept has shown some efficacy in a case study of PBC associated with rheumatoid arthritis[44]. Since ERG expression can be downregulated by inflammatory agents, it is likely that inflammation plays a key role in the loss of ERG expression in these patients. Interestingly, etanercept could restore ERG expression and reduce SMAD3 activity in the in vivo model of CCL₄-induced liver fibrosis in wild-type mice, but was ineffective in $Erg^{cEC\text{-}het}$ mice, suggesting that anti-TNF therapies exert a beneficial effect by restoring endothelial homoeostasis through the ERG pathway.

In conclusion, our findings identify ERG as a novel EC-specific regulator of canonical TGFβ/BMP, promoting EC-dependent tissue homoeostasis. We provide evidence that loss of ERG expression is an early, causative event during liver fibrogenesis. We describe EndMT in human liver disease, which showed

pathological differences between ALD, PBC and NASH patients. Therefore, we propose that ERG may represent a sensitive tissue biomarker to monitor vascular dysfunction and influence therapeutic strategies.

## Methods

**Cell culture**. Primary HUVEC were collected from umbilical cords using 0.5 mg ml⁻¹ Collagenase dissolved in HBSS warmed to 37 °C. Isolated cells were cultured in M199 media supplemented with Endothelial cell growth supplement (Sigma) and 20% bovine calf serum[22]. Hepatic sinusoidal endothelial cells (HSEC) were isolated from ~30 g human liver tissue obtained from explanted livers collected from patients in the Liver Unit at Queen Elizabeth Hospital in Birmingham with informed consent and ethics committee approval. HSEC were then cultured in medium composed of human endothelial basal growth medium (Invitrogen), 10% AB human serum (HD Supplies), 10 ng ml⁻¹ vascular endothelial growth factor (VEGF), and 10 ng ml⁻¹ hepatocyte growth factor (HGF) (PeproTech). Human ERG expression was repressed by transfection of cells with 20 nM siRNA against ERG exon 6 (Qiagen; 5′-CAGATCCTACGGCTATGGAGTA-3′) or a second siRNA (#2) targeting exon 7 (Invitrogen; 5′-ACTCTCCACGGGTTAATGCATGCTAG-3′)

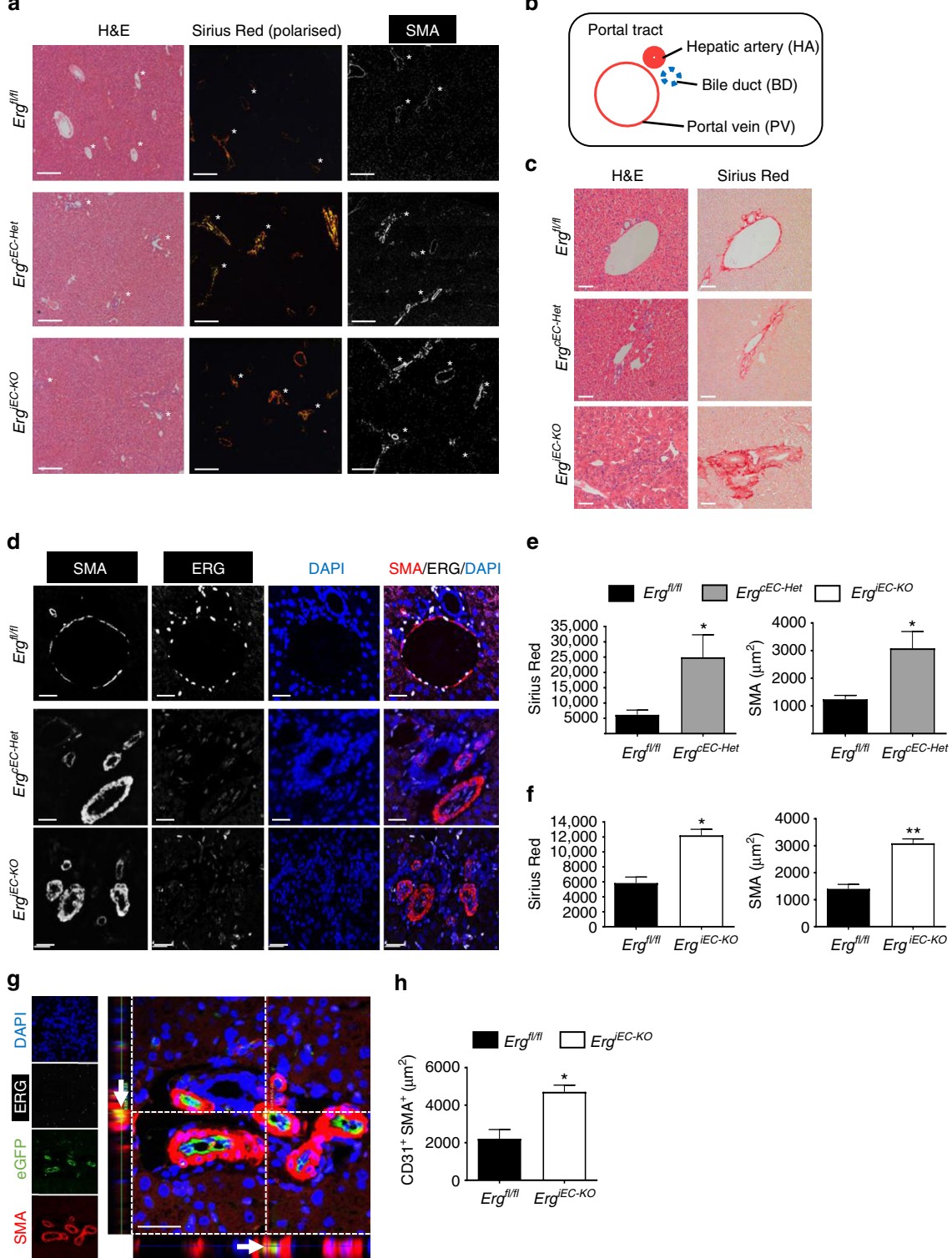

**Fig. 4** ERG-deficient mouse displays spontaneous liver fibrogenesis surrounding portal tracts. **a** EC-specific constitutive *Erg* hemi-deficient (*Erg*^cEC-het^) and inducible PDGFB-eGFP-Cre ERG flox homozygous (*Erg*^iEC-KO^) mice aged between 8 and 10 weeks were imaged by H&E, picro-sirius red and SMA (Scale bar 500 μm). Portal tract regions are identified by white asterisk with components depicted schematically in **b**. Portal tracts were assessed by **c** H&E and picro-sirius (red) (Scale bar 20 μm). **d** Immunofluorescence for SMA (grey scale; red in merge), ERG (white), DAPI (blue) and merged (Scale bar 20 μm). Images captured from **e** *Erg*^cEC-het^ and **f** *Erg*^iEC-KO^ mice were quantified for areas of Picro-sirius red positive tissue, using polarised light, and SMA expression (3 fields per mouse, *n* = 3). Scale bar 50 μm. **g** Co-localisation of SMA and eGFP expression, indicative of EndMT, was observed in *Erg*^iEC-KO^ mice by immunofluorescence (double positive cell indicated by arrow in cross-section). **h** Quantification of CD31^+^SMA^+^ double positive cells (three fields per mouse, *n* = 3–4). Data were compared to *Erg*^fl/fl^ littermate controls (*) by unpaired *t*-test. All graphical data are mean ± s.e.m., *P < 0.05, **P < 0.01

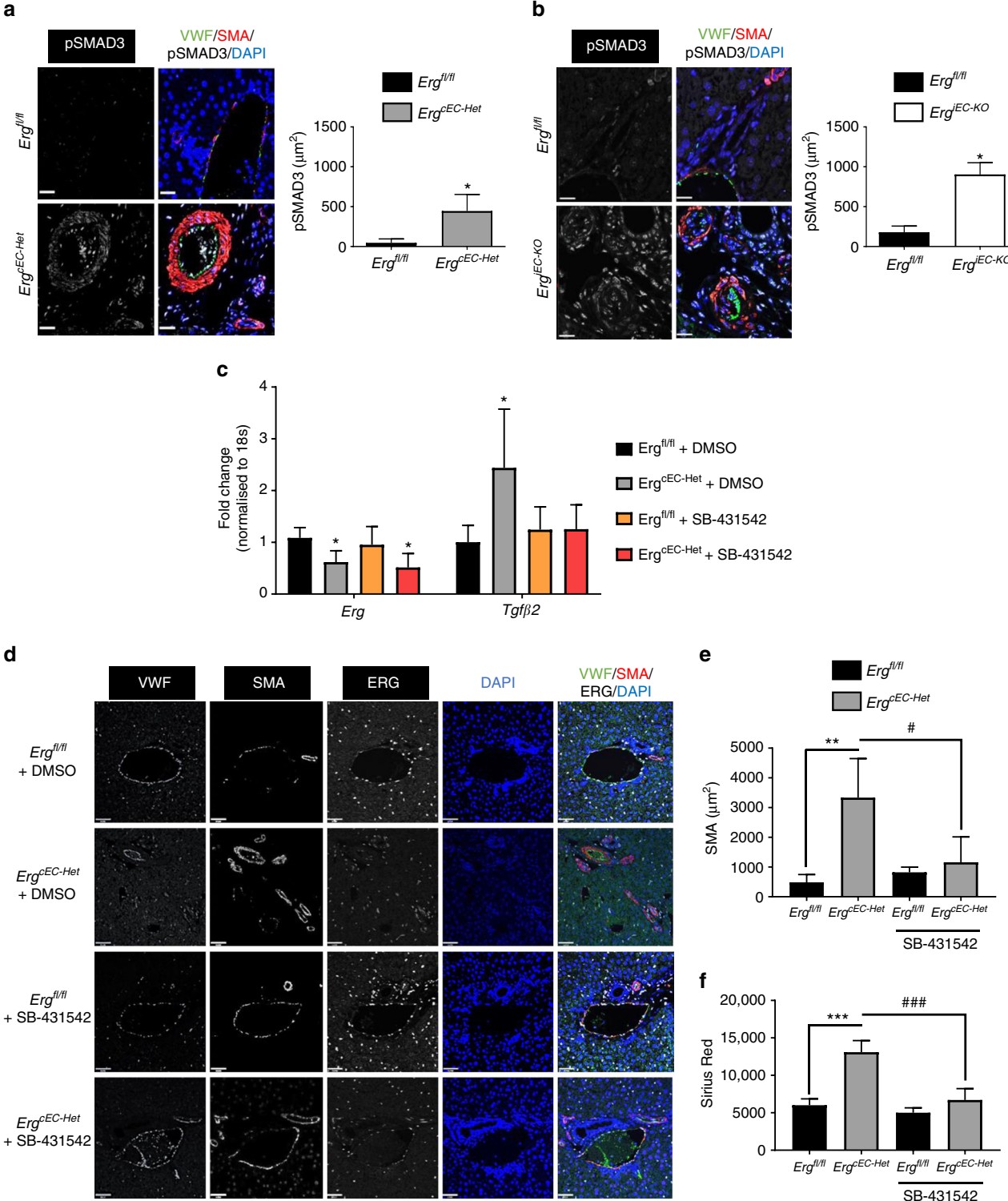

**Fig. 5** Peri-portal ERG-deficient mouse fibrogenesis phenotype is SMAD3-dependent. **a**, **b** SMAD3 activity in portal tract regions was assess by immunofluorescence for pSMAD3 (white), VWF (green), SMA (red) and DAPI (blue) in *Erg*$^{cEC-het}$ and *Erg*$^{iEC-KO}$ mice, respectively, aged 8–10 weeks. Quantification of pSMAD3 area (three fields per mouse, *n* = 3). SB-431542 (10 mg kg$^{-1}$, i.p) was administered in *Erg*$^{cEC-Het}$ mice three times a week for 2 weeks. Scale bar 20 μm. **c** mRNA was isolated from CD31$^{+}$ murine EC isolated from lung tissue and analysed by qPCR (*n* = 4). Data were compared to EC from *Erg*$^{fl/fl}$ littermate controls (*) or to DMSO treated *Erg*$^{cEC-het}$ (#) by one-way ANOVA and Bonferroni multiple comparison post-test. **d** Representative images of portal tract VWF (grey scale; green in merge), SMA (grey scale; red in merge), ERG (white), DAPI (blue) and merged panels (Scale bar 50 μm). **e** Quantification of SMA expression (5 fields per mouse, *n* = 4). **f** Quantitative analysis of Picro-sirius red positive tissue, using polarised light (more representative images in Supplementary Fig. 8B). Data were compared to *Erg*$^{fl/fl}$ littermate controls (*) or to DMSO treated *Erg*$^{cEC-het}$ (#) by one-way ANOVA and Bonferroni multiple comparison post-test. All graphical data are mean ± s.e.m., * or $^{#}P < 0.05$, ** or $^{##}P < 0.01$, *** or $^{###}P < 0.001$

of the ERG locus, both are denoted in the text as ERG siRNA. In parallel, an AllStars Negative Control siRNA (Qiagen) was used.

**Pharmacological growth factor in vitro cell treatments**. SB-431542 was purchased from Sigma (UK). SMAD3-dependent SMAD-Binding Element (SBE)

reporter (SBE4-Luc) was a gift from Bert Vogelstein (Addgene plasmid #16495)[30]. pBV-Luc empty vector, lacking a promoter sequence, was a gift from Bert Vogelstein (Addgene plasmid #16539) and was used as a control[45]. BMP response element (BRE) luciferase reporter (pGL3-BRE Luciferase) was a gift from Martine Roussel and Peter ten Dijke (Addgene plasmid #45126)[29]. Human ERG cDNA

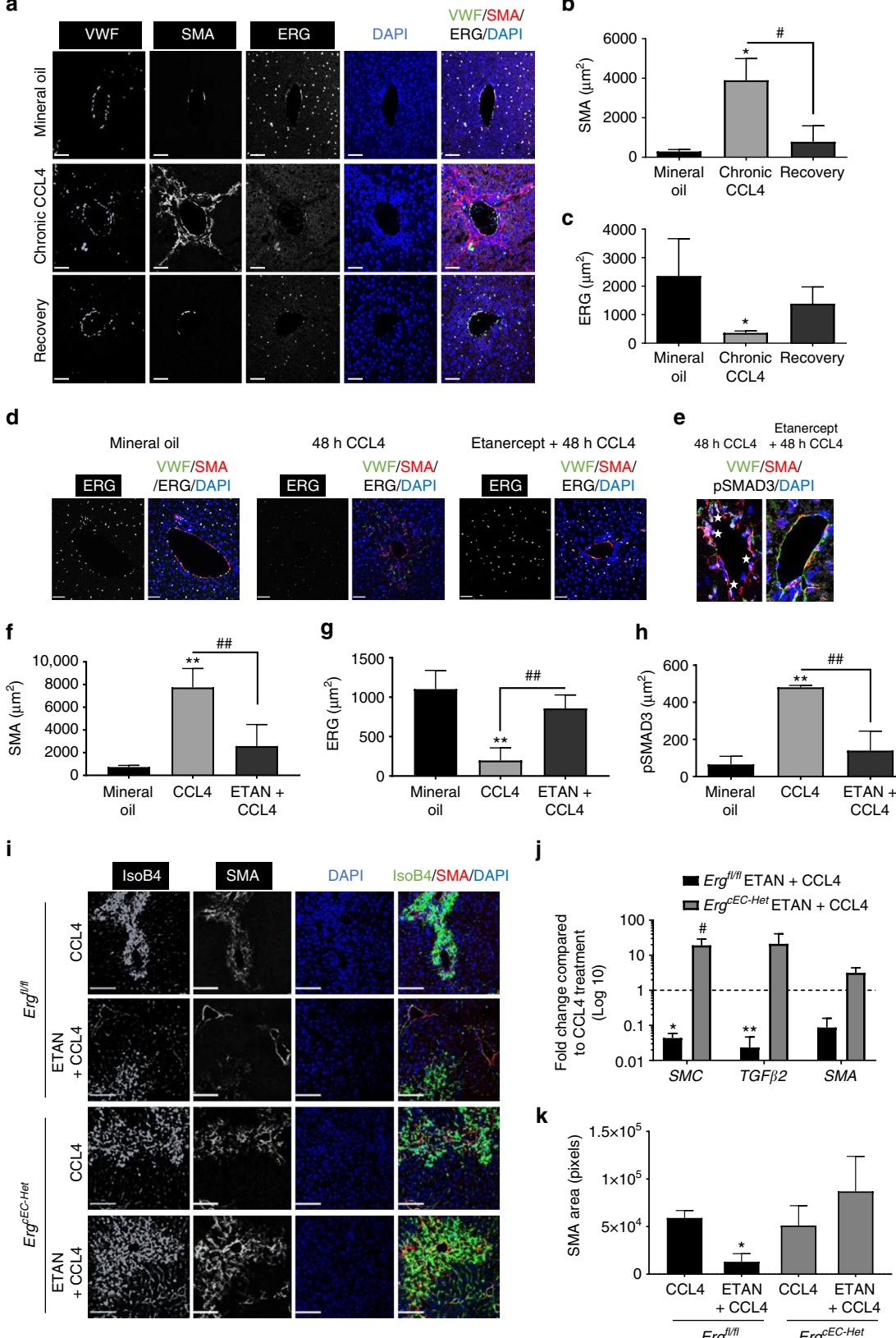

(NCBI Accession NM_182918) was cloned into the mammalian expression vector pcDNA3.1 (Invitrogen). pGL4.10[luc2] (Promega, Madison, USA) Firefly Luciferase empty vector, lacking a promoter sequence, was used as a control. pGL4.73 [hRluc/SV40] (Promega) Renilla luciferase vector was used as an internal normalisation control in the luciferase assay. HUVEC were transfected with Genejuice transfection reagent (Merck Chemicals), as recommended. Cells were incubated with 9 μl of GeneJuice, 1 μg luciferase plasmid and/or 1 μg of expression plasmid and 1 μg of pGL4-Renilla for 24 h.

**Human liver tissue sections**. Fibrotic human liver tissue was collected from patients undergoing liver transplantation surgery at the Queen Elizabeth Hospital in Birmingham for end-stage liver diseases including non-alcoholic steatohepatitis (NASH), primary biliary cholangitis (PBC) and alcohol related liver disease (ALD). Non-fibrotic tissues were used as control. All tissues were accessed with informed written patient consent and research ethics committee approval provided by the Human Biomaterials Resource Centre at the University of Birmingham, UK. Tissue was formalin fixed upon collection from patients and paraffin embedded for immunohistochemical analysis.

**Mice and breeding**. A Cre/LoxP strategy was used to develop a constitutive EC-specific heterozygous deletion of Erg using $Tie2$-$Cre$-$Erg^{fl/+}$ ($Erg^{cEC-Het}$)[23] or an inducible homozygous deletion of Erg using $Pdgfb$-$iCreER$-$eGFP/Erg^{fl/fl}$ mice ($Erg^{iEC-KO}$). All experiments with Erg-deficient mice were conducted with age and gender matched animals at Imperial College London in accordance with the UK Animals (Scientific Procedures) act of 1986. All animals used were retained on a C57BL/6 background. Both male and female mice were used for experiments and were 8–10-weeks old. All experiments were conducted using littermate controls denoted in the text as $Erg^{fl/fl}$.

**Isolation of mouse lung endothelial cells**. Primary mouse lung endothelial cells were isolated from the lungs of control $Erg^{fl/fl}$ and $Erg^{cEC-het}$ mice. Lungs were dissociated using GentleMACS C tubes and GentleMACS Dissociator (Miltenyi Biotec), digested with 0.1% collagenase type I (Invitrogen), and sieved through a 70 μm-pore cell strainer (BD Falcon). Magnetic immunosorting (Dynabeads; Invitrogen) was used to select EC by a negative sort for FcγRII/III receptor–positive macrophages followed by a positive sort for CD31–positive endothelial cells. Cells were plated in flasks precoated with a mixture of 0.1% gelatin (Sigma) and human plasma fibronectin (Chemicon) and cultured with EGM-2 media (Lonza).

**Carbon Tetrachloride CCL$_4$ liver injury model**. C57BL/6 J mice were obtained from Harlan OLAC LTD or from in-house colonies. Acute hepatic inflammation was induced using intraperitoneal injections of CCl$_4$ (carbon tetrachloride) (Sigma-Aldrich). CCl$_4$ was diluted 1:4 with mineral oil, and injected intraperitoneally (i.p) at a concentration of 1 ml kg$^{-1}$ (control animals were treated with i.p mineral oil alone). For chronic models, mice were injected with CCl$_4$ IP bi-weekly for 8 weeks. Indicated mice were allowed to recover for a further 3 weeks before the livers were collected. Experiments requiring TNF-α blockade were conducted using etanercept (10 mg kg$^{-1}$) (Amgen). Mice were injected with etanercept intravenously (i.v.) 30 min prior to single CCl$_4$ administration, control animals were given PBS alone and liver tissues collected 48 h post-injection. All CCL$_4$ experiments were conducted at University of Birmingham with experimental procedures undertaken in 6–8-week-old mice and used under procedure in accordance the Animals (Scientific Procedures) Act of 1986.

**Immunofluorescence analysis of tissue and HUVEC**. Tissues were fixed in 4% paraformaldehyde for 2 h at room temperature, then transferred to 70% ethanol prior to being embedded in paraffin. Sections were de-paraffinised with Histoclear, re-hydrated for hematoxylin and eosin and picro-sirius red or heated to 95 °C for 10 min with DAKO antigen retrieval solution for immunofluorescence staining. All

sections were blocked with 2% BSA prior to incubation with primary antibodies (listed in Supplementary Table 2).

HUVEC for immunofluorescence and proximity ligation assay (PLA) were grown on gelatinised coverslips in 12-well plates. Cells were fixed with 4% paraformaldehyde for 15 min prior to blocking with 3% BSA for 1 h. PLA was performed by manufacturer's instructions (Duolink Sigma). Primary antibodies listed in Supplementary Table 2. All tissue sections and HUVEC were mounted with Prolong Diamond (Molecular Probes) and allowed to dry overnight before imaging. Confocal microscopy was carried out on a Carl Zeiss LSM780. Images were analysed with ImageJ (NIH) and Volocity software (PerkinElmer).

**Immunoblotting analysis**. CelLytic reagent (Sigma), supplemented with 1 mM phenylmethylsufony fluoride (PMSF) protease inhibitor cocktail and phosphatase inhibitor cocktail 2 and 3 (Sigma), was used to obtain whole cell protein lysates. Immunoblots were probed with the primary antibodies detailed in Supplementary Table 2. Primary antibodies were detected using fluorescently labelled secondary antibodies: goat α-rabbit IgG DyLight 680 and goat α-mouse IgG Dylight 800 (Thermo Scientific). Odyssey CLx imaging system (LI-COR Biosciences) and Odyssey 2.1 software was used for detection and quantification of fluorescence intensity. HRP-conjugated secondary antibodies were used in some instances for chemiluminescence detection and protein levels were quantified by densitometry and normalised against loading controls. Supplementary Fig. 12 for the uncropped immunoblots.

**Immunoprecipitation (Co-IP)**. HUVECs or HSEC were treated with TGFβ2 (Human TGF-β 2; Peprotech) 10 ng ml$^{-1}$ for 30 min or SB-431542 (Abcam) 10 μg ml$^{-1}$ for 1 h after 4 h serum starvation. Cells were collected in lysis buffer (50 mM Tris, pH 7.5, 150 mM NaCl, 0.1% Igepal, 1 mM EDTA, 0.25% sodium deoxycholate)[37] supplemented with 1 mM phenylmethylsufony fluoride (PMSF), protease inhibitor cocktail, and phosphatase inhibitor cocktail 2 and 3 (Sigma). Endonuclease treatment was conducted with the addition of 500 units of Benzonase (Sigma-Aldrich, UK) for 1 h at room temperature. After clarification by centrifugation at 4500×g, 1 mg of total protein cell lysate was incubated with 3 μg of α-ERG antibody (sc-376293, Santa Cruz) or 3 μg of α-mouse IgG (eBioscience) and Protein A/G PLUS-Agarose (sc-2003, Santa Cruz Biotechnology) overnight at 4 °C. Beads with the immune-precipitated complex were washed with lysis buffer three times prior to denaturing in Laemmli buffer at 95 °C for 5 min. Samples were loaded on 12% acrylamide gels, transferred to nitrocellulose membranes and probed with α-ERG (ab133264, Abcam) and α- Phospho SMAD3 (ab52903, Abcam), α-SMAD3 (9523-S, Cell signalling), α-SMAD2 (86F7, Cell signalling), α-SMAD1 (9743 S, Cell Signalling) antibodies. HRP-conjugated secondary antibodies were used for chemiluminescence detection and protein levels were quantified by densitometry and normalised against loading controls. See Supplementary Fig. 13 for the uncropped immunoblots.

**Reporter assays**. Reporter assays in HUVEC were performed with the Dual-Luciferase Reporter Assay System (Promega) and a GloMax-Multi + Microplate Multimode Reader (Promega). Twenty-four hours after co-transfection with the luciferase reporters and expression plasmids and 18 h after stimulation with human recombinant BMP9 (1 ng ml$^{-1}$, Peprotech) or TGFβ2 (10 ng ml$^{-1}$, Peprotech), HUVEC were lysed and reporter assays were performed in triplicate. For siRNA experiments, cells were transfected with 20 nM ERG or control siRNAs and after 24 h transfected with the luciferase reporters and activated with the ligands. The ratio of luciferase signal to Renilla signal from each transfection was determined to control for well-to-well variation in transfection efficiency.

**Real-time polymerase chain reaction**. RNA was extracted from tissues and HUVEC using the RNeasy kit (Qiagen). Supercript III Reverse Transcriptase (Invitrogen) was used for first strand cDNA synthesis. Quantitative real-time PCR was carried out using PerfeCTa SYBR Green Fastmix (Quanta Biosciences) on a

**Fig. 6** Etanercept prevents ERG loss during murine liver injury. **a** C57/B6 mice were injected with mineral oil (vehicle; bi-weekly i.p) or with chronic CCl$_4$ administration (8 weeks, bi-weekly i.p); a third group was subjected to a 3-weeks recovery period following CCL$_4$ administration. Representative immunofluorescence of liver tissue samples stained for VWF (grey scale, green in merge), SMA (grey scale, red in merge) and ERG (white). Scale bar 50 μm. Quantification of **b** SMA expression and **c** ERG expression (three fields per mouse, n = 3). Data were compared to mineral oil controls (*) or to chronic CCL$_4$ administration (#) by one-way ANOVA and Bonferroni multiple comparison post-test. **d** Representative immunofluorescence following acute (48 h) mineral oil, CCL$_4$ or co-administration of etanercept and CCl$_4$. Staining for ERG (white); merged images showing VWF (green), SMA (red) and DAPI (blue). **e** pSMAD3 (white; stars); other colours as in **d**; scale bar 20 μm. **f–h**: Quantification of **f** SMA, **g** ERG and **h** pSMAD3 expression (three fields per mouse, n = 3). Data were compared to mineral oil controls (*) or to 48 h CCL$_4$ administration (#) by one-way ANOVA and Bonferroni multiple comparison post-test. **i** Co-administration of etanercept and CCL$_4$ for 48 h was able to significantly reduce SMA expression in $Erg^{fl/fl}$ mice while it was ineffective in $Erg^{cEC-Het}$ mice. Representative immunofluorescence of liver tissue, colour scheme as above. Scale bar 50 μm. **j** Quantification of SMAD2/3 target gene expression by qPCR in liver tissue of $Erg^{fl/fl}$ mice injected with etanercept and CCL$_4$ compared to CCL$_4$ alone (*) or $Erg^{cEC-Het}$ treated with etanercept and CCL$_4$ compared to CCL$_4$ alone (#). **k** Quantification of SMA expression (three fields per mouse, n = 3). Data analysed by one-way ANOVA and Bonferroni multiple comparison post-test. All graphical data are mean ± s.e.m., * or #P < 0.05, ** or ##P < 0.01, *** or ###P < 0.001

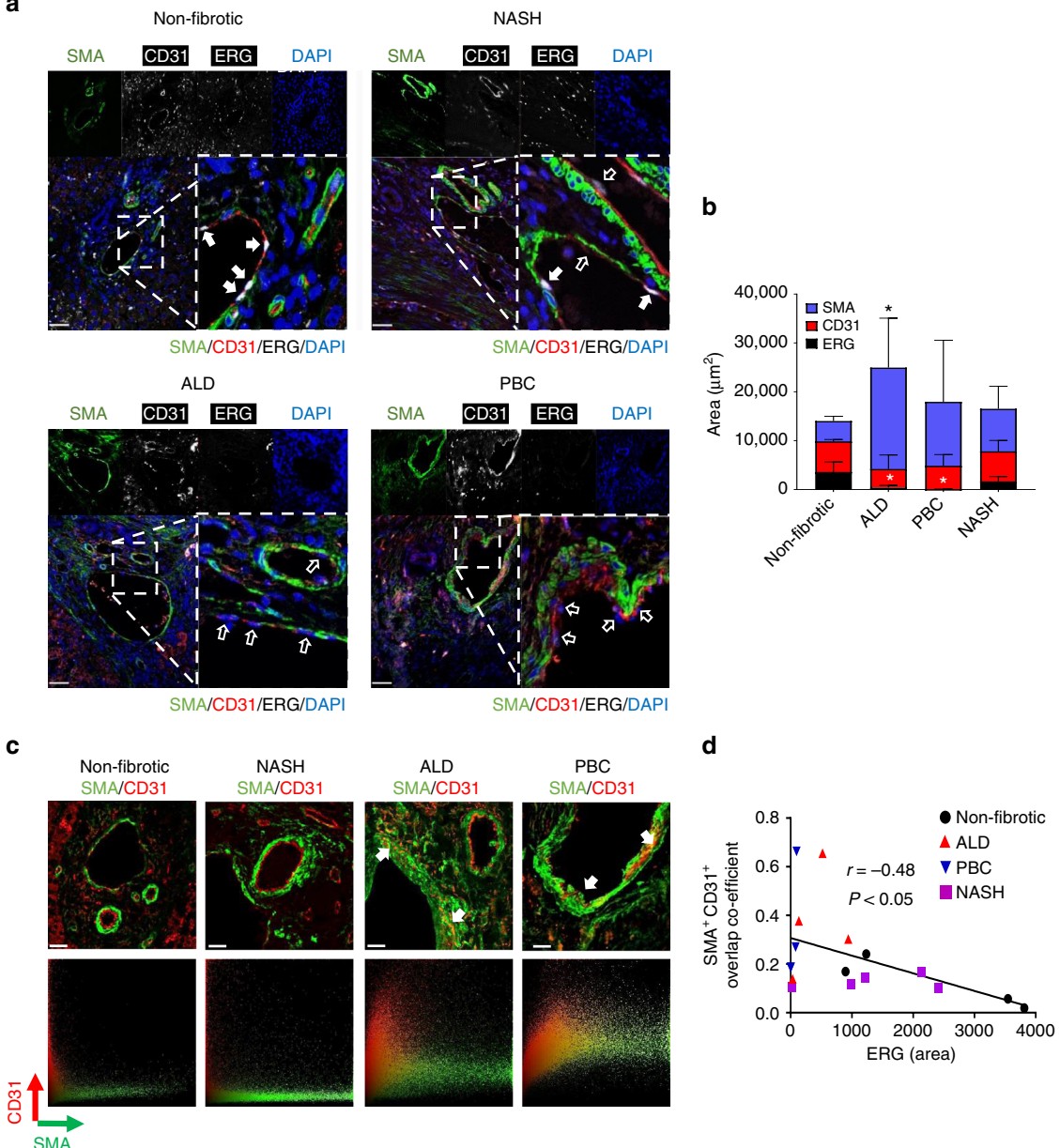

**Fig. 7** Loss of ERG expression correlates with EndMT in end-stage human liver disease. **a** Representative immunofluorescence of human liver sections stained for SMA (green), CD31 (grey scale; red in merge), ERG (white) and DAPI (blue) were captured from non-fibrotic donor patients with chronic end-stage NASH, PBC and ALD with enlarged merge panel. Filled white arrows identify CD31+ERG+ EC whereas empty white arrows show CD31+ERG− EC (Scale bar 50 μm). **b** Quantification of SMA, CD31 and ERG expression (five fields per section, n = 4–5). Scale bar 20 μm. Data were compared to non-fibrotic group (*) by one-way ANOVA and Dunnett's comparison post-test. **c** Representative images for CD31 (Red) and SMA (green) staining, showing co-localisation indicative of EndMT (arrows) and their pixel co-localisation profile in ALD, PBC and NASH patient samples. **d** Quantification of pixel co-localisation of CD31 (Red) and SMA (green) by overlap co-efficient was correlated with ERG expression for all individual human section data, non-fibrotic (black circles), NASH (purple squares), PBC (blue triangle down) and ALD (red triangle up). Correlation was calculated by linear regression, resulting in a significant negative correlation. All graphical data are mean ± s.e.m., *P < 0.05

Bio-Rad CFX96 system. Oligonucleotides used for human and mouse are listed in Supplementary Tables 3, 4, respectively.

**ChIP-qPCR.** Chromatin immunoprecipitation (ChIP) was carried out using ChIP-IT express kit (Active Motif). HUVEC were transfected with ERG or control siRNA (20 nM, for 48 h) or treated with PBS or TGFβ2 (10 ng ml⁻¹, for 30 min), and cross-linked for 10 min with formaldehyde (final concentration of 1%). Chromatin was sheared using a Bioruptor UCD-200 ultrasound sonicator (Diagenode), resulting in DNA fragments of 500–1000 bp in size. Chromatin was immunoprecipitated with 2 μg antibody to ERG (sc-354X, Santa Cruz Biotechnology), SMAD3 (#9523, Cell Signalling) or negative control rabbit IgG (PP64, Chemicon, Millipore)

using protein G magnetic beads (Active Motif). Immunoprecipitated DNA was then used as template for qPCR using primers specific for genomic loci. Oligonucleotide sequences are listed in Supplementary Table 5.

**Bioinformatics analysis.** Gene ontology analysis was performed using Database for Annotation, Visualisation, and Integrated Discovery (DAVID), to identify over-represented gene ontology categories from microarray data obtained from control and ERG-deficient HUVEC[24]. Functional clustering tool within DAVID assigned group gene associated with TGF signalling with an enrichment score of 1.3 (which corresponds to P < 0.05). Microarray data are deposited in GEO database: GSE32984[24].

The JASPAR database (http://jaspar.genereg.net) was used to identify ERG transcription factor DNA binding motifs. Data obtained from genome-wide ChIP-Seq for H3K27ac, H3K4Me3 and H3K4Me1 histone modifications and RNA polymerase II occupancy in HUVEC and phyloP sequence conservation (conservation scores between −5 and +5) were based on Multiz alignment analysis of 100 vertebrate species (derived from the Broad Institute and publicly available from the ENCODE Consortium). Each track was visualised using the UCSC Genome Browser database (https://genome.ucsc.edu/index.html).

**Statistical analysis**. Data shown are representative of at least three experiments (unless otherwise stated) and are expressed as the mean ± standard error of the mean (s.e.m.). For in vivo experiments group sizes were determined using estimates of variance and minimum detectable differences between groups that were based on our experience of characterising murine phenotypes. Animals were randomized using an identification numbers, allowing investigators to perform histological analyses blinded to animal genotype and treatment group. Statistical significance was determined by Student's unpaired $t$-test assuming unequal variances, using Prism 6.0 (Graph Pad). Differences were considered significant with a $P$-value < 0.05.

**Data availability**. The data supporting the findings of this study are available within the article and its Supplementary Information files and can also be obtained from the corresponding author upon reasonable request.

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

## Acknowledgements

Prof. S. Hubscher (University of Birmingham, UK) and Prof D. Haskard (Imperial College London, UK) for helpful discussions. Prof. H. Gerhardt (Max-Delbrück Center for Molecular Medicine, Berlin, Germany) for generation of the *Pdgfb-iCreER-eGFP/*

*Erg*<sup>fl/fl</sup> mice (*Erg*<sup>iEC-KO</sup>). The NIHR Biomedical Research Unit in Liver Disease at Birmingham University for supporting this work. This work was funded by grants from the British Heart Foundation (PG/09/096 & RG/11/17/29256). The Facility for Imaging by Light Microscopy (FILM) at Imperial College London is part supported by funding from the Wellcome Trust (grant 104931/Z/14/Z) and BBSRC (grant BB/L015129/1).

## Author contributions

N.P.D. conceived and designed and carried out in vitro and in vivo experiments, analysed and conceptualised results, wrote the manuscript. C.R.P. designed and performed experiments, analysed and interpreted results. C.R. and L.O.A. performed experiments and analysed results. Y.Y. and V.K. performed bioinfomatic analysis; A.C. and G.W. performed experiments and analysed results; G.M.B., J.C.M. and D.H.A. contributed to scientific discussion; P.L. provided reagents, advice and contributed to scientific discussion; A.M.R. provided funding, conceived, designed and supervised the study, interpreted results and wrote the manuscript.

## Additional information

**Competing interests:** The authors declare no competing financial interests.

