## [Peer Review File · Nature Communications]

Reviewers' comments:

Reviewer #1 (expert in TGFb and liver disease)

Remarks to the Author:

In the manuscript submitted by Dufton and colleagues, the authors describe how ERG interacts with SMAD1 and SMAD3 to regulate endothelial cell function and show that depletion of ERG promotes SMAD3 transcriptional activity in HUVECs and promotes liver fibrosis spontaneously and in the chronic injury model of CCl₄. In addition, decreased ERG expression is also increased in patients with liver fibrosis. The authors propose a model where ERG promotes phosphorylation of SMAD1, but does not form a complex with SMAD1, while it forms a complex with phosphorylated SMAD3 and inhibits expression of the SMAD3 target genes encoding TGF β 1 and CNN1. This study proposes an interesting hypothesis and shows that the activity of ERG in endothelial cells is a regulator of liver fibrosis, but the explanation of the underlying interaction between ERG and SMAD proteins needs to be more rigorously investigated.

The interaction between SMAD3 and ERG relies heavily on the location of SBEs. These sequences are very common in the genome, and the presence of an SBE does not indicate SMAD binding. We know that the sites of SMAD2/3 binding vary tremendously between different cell types and shift during differentiation (Kim et al, 2011 PMID: 21741376, Mullen et al., 2011 PMID: 22036565, Brown et al., 2011 PMID: 21630377, Tsankov et al, 2015 PMID: 25693565 and comparing these data to Koinuma et al., 2009 (ref 36)), so the presence of an SBE or a cluster of SBEs does not indicate that SMAD3 binds a region in a specific cell type. ChIP-PCR for SMAD3 still does not adequately address this question, because we have no idea if a 1.5 fold increase in SMAD3 binding with addition of TGF β 2 (Fig 3F and 3H) occurs as SMAD3 peak for the genes encoding TGF β 2 or CNN1. This may be statistically significant, but such a small change raises the question as to whether this is even a significant SMAD peak at either gene. SMAD3 ChIP-seq is needed to support the proposed model. The ideal experiment would be SMAD3 ChIP-seq with control siRNA and ERG siRNA (OR with and without TGF β 2 treatment as another option). This would answer the question of where SMAD3 is binding, how this binding overlaps with ERG, and whether SMAD3 is affected in the way predicted by the model with decreased ERG expression.

This paper also relies heavily on ERG ChIP-seq data, which is described as submitted. It is important to have access to this manuscript during review to understand how these data were collected and analyzed.

In Figure 3A, it is necessary to have a control. As stated above, SBEs are present throughout the genome. It may be that SBEs are enriched at the TSS of all genes expressed in HUVECs and therefore, happen to be in the promoter of many ERG-enriched genes. A comparison of the SBE density at the TSSs of ERG-enriched genes and TSSs of active genes in HUVECs should answer these questions. How does the ERG density appear on this plot? Is it similar?

SMAD3 (above references) and ERG (as shown for genes encoding SMAD1 and ACRVL1 - Supplemental Fig 3) bind enhancers as well as promoters. The analysis applied to promoters should also be addressed at enhancers, which can be defined by the H2K27Ac data already referenced in the paper.

The paper focuses on SMAD3, but makes no mention of SMAD2. Does SMAD2 play a similar role in this model? The conclusion in both culture models and in vivo models is that the ERG depletion promotes EndMT through SMAD3, but SMAD2 is not mentioned other than to show that SMAD2 mRNA expression, like SMAD3 mRNA expression does not change with depletion of ERG. SB431542 inhibits SMAD2 and SMAD3 phosphorylation, so treatment with SB431542 does not prove a phenotype is SMAD3 dependent.

In discussing the results, it is also necessary to keep in mind that ChIP data showing occupancy of the same gene or promoter (or occupancy of a promoter where there are also SBE sites) does not provide proof that the factors are binding at the same time on DNA. ChIP-re-ChIP would confirm co-occupancy, but I think this is beyond what is required as long as the data are not over interpreted.

ERG is shown to bind SMAD3 and this has been reported previously as the authors describe. It is unclear from the methods whether lysates were treated with DNase for the CO-IP experiments. This control would be helpful to show that ERG can interact with SMAD3 independent DNA as the model suggests (Supplemental Fig 6C).

Fang et al (ref 34) describes ERG as interacting with pSMAD3 to enhance transcriptional activity in the presence and absence of SB431542. Why are these results contradictory to the current findings?

Etanercept is shown to decrease CCl4 mediated induction of SMA and largely normalizes ERG mRNA levels. Is this a pathway completely independent of the ERG-SMAD3 interaction, which is the focus of this manuscript? Does etanercept affect pSMAD3 or TGFB2 or is the argument that we looking at a completely independent pathway that can also affect fibrosis? Is the activity of etanercept on SMA mediated through ERG, i.e. is the protective effect of etanercept limited in ERG fl/fl mice?

Minor comments:

Microarray data in Figure 1A shows one sample at 24 and one sample at 48 hrs. Is this one replicate or a merger of three replicates from GSE32984? We should be able to see all three replicates for each time point. This gene list also appears somewhat arbitrary in terms of the TGF and BMP genes selected. A broader list or more explanation of why these were chosen would be helpful. Figure 1A references #15 in the figure legend and #24 in the text for the source of the microarray data. I assume this reference should be #24 for both.

Fig 1 H. – the two sets of images are not labeled. Are these ERG fl/fl and ERG^{EC}-Het? What is DRAQV?

Figure 2E - Why is IgG as effective as ERG at pulling down tSMAD3?

No error bars Supplemental Figure 3.

Figure 3B- are the 715 genes occupied by ERG and containing SBEs affected with depletion of ERG? This would provide additional support that ERG binding is regulatory.
Line 126: (Fig. 43B & D)

The abbreviation EC is not defined.

Reviewer #2 (expert in TGF β /EMT/EndMT/fibrosis)
Remarks to the Author:

Dynamic regulation of canonical TGF β signaling by the endothelial transcription factor ERG protects from liver fibrogenesis.

Neil P. Dufton et al.

The authors report their original work pertaining to the role of the EC-restricted ETS-related gene (ERG) in development of chronic liver disease. They suggest that transcription factor, ERG, regulates endothelial homeostasis primarily by binding to phosphorylated Smad3 and antagonizing its transcriptional activity in mediating TGF β dependent EndoMT. Additionally, ERG binds to key TGF/BMP

activated gene loci, proximal to their transcription start sites, to positively regulate expression of ACVRL1, ENG, and SMAD1. The authors show that this regulation is required for activation of ALK1 /Smad1 signaling. Overall, therefore, they show that ERG is a pivotal regulator of the balance between TGF β -Smad3 signaling versus ALK1-Smad1 signaling in EC, to regulate endothelial stability versus EndoMT and its consequent initiation and perpetuation of chronic inflammatory processes. The manuscript utilizes molecular and cellular studies in HUVECs, as well as EC-specific heterozygous mouse gene KO and conditional KO of ERG to address this issue. Finally, they validate the relevance to human liver disease showing that in (fibrotic liver disease) alcoholic liver disease and PBC but not NASH, there is a decrease in ERG expression and elevated EC and non-EC pSmad3 expression.

Overall the study is well executed, and provides novel and important insights, into the balance between TGF beta and BMP signaling in endothelial homeostasis, which will be of interest to the vascular biology community. Moreover, the manuscript for the first time reports the existence of EndoMT in hepatic EC, and reveals that the ultimately initiates events leading to hepatic fibrosis, presumably due to loss of vascular integrity and/or paracrine activities of a defective endothelial layer. However some of the data needs better validation. This reviewer has concerns about the quality of some of the molecular data that shows interactions between ERG and Smads (see below).

Figure 1B: It may be more helpful to show decreases in expression as negative fold decreases below X axis. Figure 1C there is no clear decrease in Smad1 in the representative Western shown, thus Figure 1D does not reflect Fig 1C.

The quality of the data pertaining to Smad3 in Figure 2E is poor. Moreover, there is no positive control for pSmad3 or total Smad3, and no size markers, we therefore do not know which are the expected band(s). Since the tSmad3 bands appear identical in the non-specific IgG pulldown lane as with the anti-ERG-IgG lane, there is no evidence whatsoever for direct binding between the two proteins (ERG and pSMAD3). The complete disappearance of a tSMAD3 band in response to SB431542 treatment is in marked contrast to the minimally reduced levels of pSmad3 in the upper panel. The quality of the upper panel (pSmad3) is very poor, and a more convincing demonstration of co-PI would be needed for publication, since this is a key and important finding of the manuscript.

Similarly, the quality of the data in Supplementary Figure 4 (pSmad3) is poor. Moreover, since the major take-home message from Fig S4 is that siERG reduces levels of total and phospho-SMAD1 and increases total and pSmad3, the lanes for the two different siRNA species should be placed on the same gel, e.g. interspersed between each other at different time points.

In Figure S1. What is the evidence that only TGFbeta2 and not TGFb1 binds and activates ALK1? If this is from another reference, please quote it, as the data is not shown in the manuscript. On that point too, why is such a high concentration of TGFbeta2 used throughout the manuscript (10ng/ml)? This is excessive and might result in off-target activities due to contaminants. Have any experiments be undertaken at lower concentrations? e.g. 1ng/ml or less?

There are a few grammatical or typographical errors:

In line 126: the figure number and reference are unclear.

Line 140 "... expression of the SMAD1 gene itself, as well as ACVRL1 and ENG", and is required
Line 144-145 "...ERG may interact with SMAD3 during binding to genomic DNA. To investigate such potential interactions between ERG/Smad3 complexes, we analysed data from genome-wide ERG ChIP-seq"

Line 146 The SBE motif was present...

Line 174 of in

Line 180 SB-431542 is an Alk5 inhibitor, not a Smad3 inhibitor,

Line 299... the Smad1 pathway, including the SMAD1 gene itself

Legend to Figure 1, intensity not intensity
Legend to Figure 2 interactions not interacts

Manuscript NCOMMS-16-23645-T - Resubmission**Response to reviewers**

Reviewer #1: We thank the Reviewer for his/her comments

Comments:

1. *The interaction between SMAD3 and ERG relies heavily on the location of SBEs. These sequences are very common in the genome, and the presence of an SBE does not indicate SMAD binding. We know that the sites of SMAD2/3 binding vary tremendously between different cell types and shift during differentiation (Kim et al, 2011 PMID: 21741376, Mullen et al., 2011 PMID: 22036565, Brown et al., 2011 PMID: 21630377, Tsankov et al, 2015 PMID: 25693565 and comparing these data to Koinuma et al., 2009 (ref 36)), so the presence of an SBE or a cluster of SBEs does not indicate that SMAD3 binds a region in a specific cell type. ChIP-PCR for SMAD3 still does not adequately address this question, because we have no idea if a 1.5-fold increase in SMAD3 binding with addition of TGFβ2 (Fig 3F and 3H) occurs as SMAD3 peak for the genes encoding TGFβ2 or CNN1. This may be statistically significant, but such a small change raises the question as to whether this is even a significant SMAD peak at either gene. SMAD3 ChIP-seq is needed to support the proposed model. The ideal experiment would be SMAD3 ChIP-seq with control siRNA and ERG siRNA (OR with and without TGFβ2 treatment as another option). This would answer the question of where SMAD3 is binding, how this binding overlaps with ERG, and whether SMAD3 is affected in the way predicted by the model with decreased ERG expression.*

We agree with the Reviewer's observation that the presence of SBE motifs is not *per se* indicative of SMAD binding, and indeed we use ChIP-PCR to investigate TF binding to DNA. The Reviewer expresses concern that a statistically significant 1.5-fold increase in SMAD3 binding to DNA shown by ChIP-PCR may not be functionally relevant. The data actually show a 3-fold enrichment of SMAD3 within our regions of interest, compared to a negative control region; the 1.5-fold increase refers to TGFβ2 treatment compared to baseline. To address the Reviewer's concern, we investigated SMAD3 enrichment at a previously described SMAD3 binding locus on the PAI-1 promoter (*Fu et al., 2009*)¹. *Fu et al.* show that an increase in PAI-1 mRNA expression following TGFβ treatment is characterised by significantly enhanced SMAD3 and H3K27Ac enrichment within this region. Using the same primers detailed by *Fu et al.*, we observe a significant increase in SMAD3 enrichment at this site following TGFβ2 stimulation (see Appendix 1A), which is comparable to that observed with our investigated target genes (Fig. 3 D & F), supporting the view that these levels of enrichment are indeed functionally relevant.

The Reviewer suggests a SMAD3 ChIP-Seq study for a genome-wide analysis of the relationship between these two transcription factors. We agree this would be extremely insightful as a global comparison; however, a ChIP-Seq study would not, in our view, address the Reviewer's concerns regarding the specificity or functional significance of specific SMAD3 ChIP-PCR data. In the context of this study, we do not believe that a SMAD3 ChIP-Seq, which is a complex experiment requiring extensive and careful optimisation and data analysis, is critical to support the main findings of this study.

2. *This paper also relies heavily on ERG ChIP-seq data, which is described as submitted. It is important to have access to this manuscript during review to understand how these data were collected and analyzed. In Figure 3A, it is necessary to have a control. As stated above, SBEs are present throughout the genome. It may be that SBEs are enriched at the TSS of all genes expressed in HUVECs and therefore, happen to be in the promoter of many ERG-enriched genes. A comparison of the SBE density at the TSSs of ERG-enriched genes and TSSs of active genes in HUVECs should answer this questions. How does the ERG density appear on this plot? Is it similar?*

In discussing the results, it is also necessary to keep in mind that ChIP data showing occupancy of the same gene or promoter (or occupancy of a promoter where there are also SBE sites) does not provide proof that the factors are binding at the same time on DNA. ChIP-re-ChIP would confirm co-occupancy, but I think this is beyond what is required as long as the data are not over interpreted.

We appreciate the Reviewer's request to have access to the ERG ChIP-Seq data. However, this is part of a manuscript with a completely separate focus (*Yang et al*) currently under review in Circulation Research (already shared with Dr Todorovic, available to reviewers upon request). Since submission, we have been made aware that Nature Communication's editorial requirements do not allow reference to a manuscript under review. Thus, we have removed the ERG Chip-Seq data within the current manuscript (original Fig. 3A-D and Supplementary Fig. 3 original submission) and have replaced it with analysis of sequence conservation of ERG-DNA binding motifs between 100 vertebrates (new Figure. 3 A & B). We have modified the text on lines 144-150 and 116-118 accordingly. We hope that these modifications address the Reviewer's concerns.

3. *The paper focuses on SMAD3, but makes no mention of SMAD2. Does SMAD2 play a similar role in this model? The conclusion in both culture models and in vivo models is that the ERG depletion promotes EndMT through SMAD3, but SMAD2 is not mentioned other than to show that SMAD2 mRNA expression, like SMAD3 mRNA expression does not change with depletion of ERG. SB431542 inhibits SMAD2 and SMAD3 phosphorylation, so treatment with SB431542 does not prove a phenotype is SMAD3 dependent.*

This is a fair point, which we have addressed this both experimentally and by modifying the text where required. We tested whether ERG could form a complex with SMAD2 as well as SMAD3, and interestingly we found that ERG is also able to physically interact with SMAD2; these data are presented in new Fig. 2F.

As mentioned by this Reviewer, SB-431542 inhibits both SMAD2 and SMAD3 phosphorylation; we have corrected the text accordingly to include SMAD2 in the interpretation of the results obtained with this tool. We also attempted to define specific roles of SMAD2 and SMAD3 by using individual siRNAs (Appendix 1B and C). To confirm the role of SMAD3 in regulating EndMT-associated genes we used SMAD3 siRNA plus or minus TGFβ2 in HUVEC. In line with our model, we find that TGFβ2-induced up-regulation of TGFβ2 and CNN1 expression is SMAD3-dependent (Appendix 1B). These data support our model that SMAD3 is a crucial component in driving the EndMT transcriptional profile. However, we were unable to use the same approach to determine the role of SMAD2 in regulating EndMT-associated genes, because the SMAD2 siRNA also regulated ERG expression (Appendix 1C), making it impossible to determine if the role of SMAD2 in EndMT is ERG-dependent.

Since we cannot definitively assign specificity for our phenotype to SMAD3 alone, we have modified the manuscript to capture the potential for both SMAD2/3 to play a role within the proposed mechanism of action. However, given that SMAD2 does not directly bind to DNA², the investigation of DNA binding by ChIP-PCR remains focused on SMAD3.

4. *ERG is shown to bind SMAD3 and this has been reported previously as the authors describe. It is unclear from the methods whether lysates were treated with DNase for the CO-IP experiments. This control would be helpful to show that ERG can interact with SMAD3 independent DNA as the model suggests (Supplemental Fig 6C).*

Apologies for the lack of clarity in the methods: DNase was not used in the co-IP experiments. To investigate whether DNA was required to preserve this interaction, whole protein lysates were treated with Benzonase, a potent endonuclease that has no proteolytic activity. We show that incubation of 500 units of Benzonase for 1 h was sufficient to remove DNA from our protein lysate; however this did not affect the integrity of ERG-pSMAD3 or tSMAD3 complex (new Supplementary Fig. 5 A & B). These data also support the model of ERG-SMAD2/3 interaction (original Supplementary Fig. 6C) which has been modified for clarity (new Supplementary Fig. 6C).

5. Fang et al (ref 34) describes ERG as interacting with pSMAD3 to enhance transcriptional activity in the presence and absence of SB431542. Why are these results contradictory to the current findings?

There is evidence that ERG plays a different role depending on cell lineage; this is indeed the focus of current studies in our group. Whilst in endothelial cells ERG has a homeostatic function, in prostate cancer ERG acts as an oncogene when aberrantly expressed in epithelial cells as a result of a chromosomal translocation (TMPRSS2:ERG fusion). It is notable that the Fang et al. paper describes the interaction between ERG and SMAD3 by over-expressing both proteins in a prostate cancer cell line. Our analysis of global transcriptome and cistrome data comparing ERG microarray and ChIP-seq in HUVEC with ERG microarray and ChIP-seq in prostate cancer cells (Yang et al., Circ Res under review; see response 2), reveals that <10% of ERG targets are shared between the two cell types. Therefore, these data are in line with the model that the functional outcome of ERG and SMAD3 interaction may be dependent on the cell type.

6. Etanercept is shown to decrease CCL4 mediated induction of SMA and largely normalizes ERG mRNA levels. Is this a pathway completely independent of the ERG-SMAD3 interaction, which is the focus of this manuscript? Does etanercept affect pSMAD3 or TGFB2 or is the argument that we looking at a completely independent pathway that can also affect fibrosis? Is the activity of etanercept on SMA mediated through ERG, i.e. is the protective effect of etanercept limited in ERG fl/fl mice?

We proposed that the mechanism through which etanercept protects from CCL4-mediated liver injury is directly connected to the ERG-SMAD3 pathway; CCL4 down regulates ERG and promotes SMAD3-dependent gene expression which was normalised by etanercept. We addressed the reviewer's questions of the effect of etanercept on SMAD3 activity in the CCL4 model. We show that acute and chronic CCL4 administration resulted in SMAD3 phosphorylation in liver tissue (new Supplementary Fig. 9B and C); co-administration of etanercept and CCL4 significantly reduced SMAD3 phosphorylation in liver tissue (new Fig. 6E & H and Supplementary Figure 9C).

To address the question raised by this Reviewer on whether the protective effect of etanercept depends on ERG, we carried out new experiments using *Erg* hemi-deficient mice (*Erg*^{CEC-Het}). These showed that the protective effects of etanercept on CCL4-induced SMA expression (new Fig. 6I & K) and gene transcription (new Fig. 6J) within whole liver tissue are mediated by ERG, since etanercept did not correct the CCL4 - induced phenotypes in *Erg*^{CEC-Het} mice.

Minor Comments

Microarray data in Figure 1A shows one sample at 24 and one sample at 48 hrs. Is this one replicate or a merger of three replicates from GSE32984? ... This gene list also appears somewhat arbitrary in terms of the TGF and BMP genes selected. A broader list or more explanation of why these were chosen would be helpful. Figure 1A references #15 in the figure legend and #24 in the text for the source of the microarray data. I assume this reference should be #24 for both.

The microarray study was performed in triplicate, using ERG antisense at 24 hr and 48 hr; the data from 3 biological replicates was pooled for each time point (details are provided in Birdsey et al, 2012³). The list of TGFβ and BMP-related genes were obtained by DAVID gene ontology analysis of the putative ERG targets from the microarray study. This identified 41 genes as significantly enriched ($P < 0.05$). We have now included this list in new Supplementary Table 1. We apologise for our oversight in the references; #15 has been corrected to #24.

Fig 1 H. – the two sets of images are not labeled. Are these ERG fl/fl and ERG^{CEC-Het}? What is DRAQV?

We apologise for the lack of labels which have been added to denote genotype. DRAQV is a far red dye that incorporates into DNA in the same manner as DAPI.

Figure 2E - Why is IgG as effective as ERG at pulling down tSMAD3?

We accept that the quality of the Co-IP was not satisfactory, and have therefore repeated the experiments, in HUVEC and in hepatic EC. The new Fig. 2F clearly shows low background in the IgG sample compared to the anti-ERG sample. Reviewer 2 raised several points about the IP, so please refer to the reply to Reviewer 2 for more details.

No error bars Supplemental Figure 3.

These data have now been removed please refer to Response 2.

Figure 3B- are the 715 genes occupied by ERG and containing SBEs affected with depletion of ERG? This would provide additional support that ERG binding is regulatory.

Please refer to Response 2 concerning ERG Chip-Seq data, which has now been removed.

Line 126: (Fig. 43B & D)

Corrected line 125-127

The abbreviation EC is not defined.

Defined on line 46

Reviewer #2: We thank the Reviewer for his/her comments

- 1. Figure 1B: It may be more helpful to show decreases in expression as negative fold decreases below X axis. Figure 1C there is no clear decrease in Smad1 in the representative Western shown, thus Figure 1D does not reflect Fig 1C.*

We thank the reviewer for the useful suggestions. Figure 1B has been re-drawn using log₂ fold change as requested. Additional western blotting of SMAD1 was performed in Control and ERG siRNA transfected HUVEC (new Supplementary Fig. 2C), and the new Fig 1C better reflects the ~50% reduction of SMAD1 in ERG-deficient cells, as shown by the quantification in Fig. 1D.

- 2. The quality of the data pertaining to Smad3 in Figure 2E is poor. Moreover, there is no positive control for pSmad3 or total Smad3, and no size markers, we therefore do not know which are the expected band(s). Since the tSmad3 bands appear identical in the non-specific IgG pulldown lane as with the anti-ERG-IgG lane, there is no evidence whatsoever for direct binding between the two proteins (ERG and pSMAD3). The complete disappearance of a tSMAD3 band in response to SB431542 treatment is in marked contrast to the minimally reduced levels of pSmad3 in the upper panel. The quality of the upper panel (pSmad3) is very poor, and a more convincing demonstration of Co-IP would be needed for publication, since this is a key and important finding of the manuscript.*

We accept that the quality of these data could be improved. We have addressed these concerns by re-optimising the IP protocol and we have then carried out new experiments. We utilised the lysis and wash buffer described by Fang et al⁴, (detailed in Methods section lines 374-379) and used a larger amount of protein for the pull-down. In line with our initial observations, we now show clearly by Co-IP (new Fig 2F) and PLA (new Fig. 2H) that ERG binds total and phospho-SMAD3 in HUVEC. In these experiments the discrepancy between total and phospho-SMAD3 following SB-431542 treatment is no longer apparent. Crucially, we show that this interaction also occurs in human liver sinusoidal EC (HSEC, new Fig 2G). Furthermore, we show that ERG can also bind SMAD2 (new Fig. 2F), providing evidence that ERG can form a regulatory complex with SMAD2 and SMAD3. Moreover, we use Co-IP (new Supplementary Fig. 4A) and PLA (new Supplementary Fig. 4D) to show that ERG does not interact with SMAD1 in HUVEC and HSEC.

As requested, we have also provided positive control samples for total cell lysate collected prior to Co-IP as well as input and unbound fractions. All gels have molecular weight markers to ensure the correct bands could be identified by size.

3. *Similarly, the quality of the data in Supplementary Figure 4 (pSmad3) is poor. Moreover, since the major take-home message from Fig S4 is that siERG reduces levels of total and phospho-SMAD1 and increases total and pSmad3, the lanes for the two different siRNA species should be placed on the same gel, e.g. interspersed between each other at different time points.*

We followed the Reviewer's suggestions by repeating the western blots from the original supplementary figure 4, interspersing Control and ERG siRNA-treated HUVEC samples (new Supplementary Fig. 3). These additional experiments profiling TGFβ2-induced phosphorylation of SMAD1 and SMAD3 resulted in reduced SMAD1 activation and comparable response SMAD3 activation. These data are consistent with our original observations. Notably, we found a significant increase in pSMAD3 in basal conditions ERG-deficient HUVEC (new Supplementary Fig. 3F) which was confirmed by immunofluorescence (new Supplementary Fig. 3G).

4. *In Figure S1. What is the evidence that only TGFbeta2 and not TGFb1 binds and activates ALK1? If this is from another reference, please quote it, as the data is not shown in the manuscript. On that point too, why is such a high concentration of TGFbeta2 used throughout the manuscript (10ng/ml)? This is excessive and might result in off-target activities due to contaminants. Have any experiments be undertaken at lower concentrations? e.g. 1ng/ml or less?*

We apologise for the ambiguity of the schematic in Supplementary Fig. 1. We have modified the figure to capture TGFβ1 and 2 activation of canonical TGF/BMP signalling^{5,6}.

With respect to the concern about off-target effects due to contaminants, we had previously carried out experiments with lower concentrations of TGFβ2, and found comparable results. We have now included these data in new Supplementary Fig 3A, which shows 1 ng ml⁻¹ and 10 ng ml⁻¹ were equally capable of inducing CNN1 expression and SBE luciferase reporter activity. Furthermore, this is supported by a TGFβ2 concentration-response curve showing that both 1 ng ml⁻¹ and 10 ng ml⁻¹ significantly induce CNN1 gene expression (Appendix 2). Evidence of ligand specificity for BMP9 (1 ng ml⁻¹) and TGFβ2 (10 ng ml⁻¹) is provided by the luciferase assay (Fig. 2A & C) and transcriptional analysis, which show no cross-activation of canonical signalling (new Supplementary Fig. 3B). These data indicate that these concentrations can specifically activate their respective pathways. Furthermore, our results with TGFβ2 (10ng ml⁻¹) are in line with concentrations used in driving EndMT gene expression in previous studies^{7,8}.

There are a few grammatical or typographical errors

Apologies for these, all have been corrected.

Appendix 1

A

SMAD3 ChIP

B

C

(A) ChIP PCR data for SMAD3 binding to PAI1 promoter region in HUVEC. Cells were treated with either PBS or TGFβ2 (10 ng ml⁻¹) for 30 min. (B) HUVEC were transfected with Control (black) or SMAD3 siRNA (red) for 24 h prior to treatment with either PBS or TGFβ2 (10 ng ml⁻¹) for 24 hours. Expression of target genes *TGFβ2* and *CNN1* was assessed by qPCR. (C) HUVEC were transfected with Control (dash line), SMAD2 (blue) or SMAD3 (red) siRNA for 24 h. Expression of SMAD2 (left panel), SMAD3 (middle panel) and ERG (right panel) was assessed by qPCR. All graphical data are mean ± s.e.m, *P < 0.05, **P < 0.01, ***P < 0.001 was normalized to GAPDH and compared to control siRNA treated by unpaired T-test.

Appendix 2

Induction of CNN1 expression in HUVEC in response to increasing concentrations of TGFβ2. Data normalized to GAPDH and compared to PBS treated by unpaired T-test. Graphical data are mean ± s.e.m, *P < 0.05.

Reference List

1. Fu,Y. *et al.* Differential regulation of transforming growth factor beta signaling pathways by Notch in human endothelial cells. *J. Biol. Chem.* **284**, 19452-19462 (2009).
2. Yagi,K. *et al.* Alternatively spliced variant of Smad2 lacking exon 3. Comparison with wild-type Smad2 and Smad3. *J. Biol. Chem.* **274**, 703-709 (1999).
3. Birdsey,G.M. *et al.* The transcription factor Erg regulates expression of histone deacetylase 6 and multiple pathways involved in endothelial cell migration and angiogenesis. *Blood* **119**, 894-903 (2012).
4. Fang,J. *et al.* Ets Related Gene and Smad3 Proteins Collaborate to Activate Transforming Growth Factor-Beta Mediated Signaling Pathway in ETS Related Gene-Positive Prostate Cancer Cells. *J. Pharm. Sci. Pharmacol.* **1**, 175-181 (2014).
5. Lebrin,F., Deckers,M., Bertolino,P., & Ten,D.P. TGF-beta receptor function in the endothelium. *Cardiovasc. Res.* **65**, 599-608 (2005).
6. Pardali,E., Goumans,M.J., & Ten,D.P. Signaling by members of the TGF-beta family in vascular morphogenesis and disease. *Trends Cell Biol.* **20**, 556-567 (2010).
7. Medici,D. *et al.* Conversion of vascular endothelial cells into multipotent stem-like cells. *Nat. Med.* **16**, 1400-1406 (2010).
8. Xiao,L. *et al.* Tumor Endothelial Cells with Distinct Patterns of TGFbeta-Driven Endothelial-to-Mesenchymal Transition. *Cancer Res.* **75**, 1244-1254 (2015).

REVIEWERS' COMMENTS:

Reviewer #1 (Remarks to the Author):

The authors have addressed all the key issue with their resubmission.

One minor comment: Figure legend 3 refers to grey bars being putative ERG binding sites. Do the black bars represent putative Smad binding elements?

Reviewer #2 (Remarks to the Author):

Dynamic regulation of canonical TGF β signaling by the endothelial transcription factor ERG protects from liver fibrogenesis.

Neil P. Dufton et al.

The authors report their original work pertaining to the role of the EC-restricted ETS-related gene (ERG) in development of chronic liver disease. They suggest that transcription factor, ERG, regulates endothelial homeostasis primarily by binding to phosphorylated Smad3 and antagonizing its transcriptional activity in mediating TGF β dependent EndoMT. Additionally, ERG binds to key TGF/BMP activated gene loci, proximal to their transcription start sites, to positively regulate expression of ACVRL1, ENG, and SMAD1. The authors show that this regulation is required for efficient activation of ALK1 /Smad1 signaling. Overall, therefore, they show that ERG is a pivotal regulator of the balance between TGF β -Smad3 signaling versus ALK1-Smad1 signaling in EC, to regulate endothelial stability versus EndoMT and its consequent initiation and perpetuation of chronic inflammatory processes. The manuscript utilizes molecular and cellular studies in HUVECs, as well as EC-specific heterozygous mouse gene KOs and conditional KOs of ERG to address this issue. Finally, they validate the relevance to human liver disease showing that in (fibrotic liver disease) alcoholic liver disease and PBC but not NASH, there is a decrease in ERG expression and elevated EC and non-EC pSmad3 expression.

Overall the study is well executed, and provides novel and important insights, into the balance between TGF beta and BMP signaling in endothelial homeostasis, which will be of interest to the vascular biology community. Moreover, the manuscript for the first time reports the existence of EndoMT in hepatic EC, and reveals that the ultimately initiates events leading to hepatic fibrosis, presumably due to loss of vascular integrity and/or paracrine activities of a defective endothelial layer.

The authors have adequately addressed the reviewers' questions and additionally shown, using Erg^{-/-} mice that the action of etanercept in suppressing CCl₄-induced pSmad3-driven liver fibrosis is dependent on ERG.

The Figure however need attention before publication. Figures need legible labels e.g. axes on graphs, using consistent font style and size throughout as per the nature instructions.

Figure 1C right panel – which protein is assayed? Figure 1E what are the white arrows pointing to- are these supposedly EC-specific nuclei, versus the rest of the staining that is hepatocyte? Please clarify.

Manuscript NCOMMS-16-23645-T - Resubmission

Response to reviewers

Reviewer #1: We thank the Reviewer for his/her comments

Comments:

1. *One minor comment: Figure legend 3 refers to grey bars being putative ERG binding sites. Do the black bars represent putative Smad binding elements?*

The schematic diagrams only visualise putative ERG binding sites with grey bars. We have lightened the appearance of darker bars to avoid ambiguity.

Reviewer #2: We thank the Reviewer for his/her comments

1. *The Figure however need attention before publication. Figures need legible labels e.g. axes on graphs, using consistent font style and size throughout as per the nature instructions. Figure 1C right panel – which protein is assayed?*

We have addressed the formatting of the figures to ensure consistency. The right panel in Figure 1C is validating the efficacy of ERG siRNA – for clarity, we have moved this graph to figure 1D together with the quantification of the other proteins investigated.

2. *Figure 1E what are the white arrows pointing to- are these supposedly EC-specific nuclei, versus the rest of the staining that is hepatocyte?*

The white arrows highlight that the expression of SMAD1 expression is most prominent in sinusoidal endothelial cells. For clarification we have added a statement within the legend for figure 1.